# CLASS-WISE DISPARITY IN ADVERSARIAL TRAINING: IMPLICIT BIAS PERSPECTIVE

## ABSTRACT

Disparities in class-wise robust accuracies frequently arise in adversarial training, where certain classes suffer significantly lower robustness than others, even when trained on balanced data. This phenomenon has been identified and termed robust fairness in prior work, highlighting the challenge of ensuring equitable robustness across classes. In this work, we investigate the root causes of such disparities and identify a strong correlation between the norms of head parameters (i.e., the last layer's weights) and class-wise robust accuracies. Our theoretical and empirical analyses show that adversarial training tends to amplify these disparities by disproportionately affecting head norms, which in turn influence class-wise performance. To address this, we propose a simple yet effective solution that mitigates these imbalances by directly fine-tuning the head parameters while keeping the feature extractor fixed. Unlike existing methods that rely on class reweighting or remargining strategies, our approach requires no validation set and introduces minimal computational overhead. Experiments across various datasets and architectures demonstrate that our method significantly reduces disparities in class-wise robust accuracies with minimal impact on average accuracy and overall robustness, providing a practical and principled step toward improving robust fairness in adversarial learning.

## 1 INTRODUCTION

Adversarial training has become one of the most effective paradigms for improving model robustness against adversarial perturbations. While considerable progress has been made in enhancing the average robustness of deep neural networks, a critical and underexplored issue has emerged: adversarially trained models often suffer from large performance disparities across classes, even when trained on class-balanced datasets.

This phenomenon manifests as certain classes (e.g., *automobile* or *ship* in CIFAR-10) achieving much higher adversarial robustness than others (e.g., *cat* or *dog*), despite no difference in class frequency. The disparity becomes especially prominent under strong adversarial attacks. This uneven distribution of robustness has been identified and termed as robust fairness in recent works [20; 3]. It refers to the class-wise imbalance in robustness that arises naturally during adversarial training, highlighting a fairness issue distinct from average accuracy or overall robustness metrics.

Importantly, this notion of robust fairness is conceptually different from the more widely studied fairness problems based on sensitive attributes such as race, gender, or age. While traditional fairness in machine learning typically addresses bias with respect to demographic subgroups, often requiring the presence of explicit group labels, robust fairness focuses on disparities across semantic classes in multi-class classification tasks. Here, each class (e.g., *cat*, *dog*, *airplane*) is treated uniformly during training, yet still experiences varying levels of vulnerability to adversarial attacks. This reveals a fundamentally different kind of fairness issue that does not rely on external group annotations, but arises intrinsically from the learning dynamics of adversarial training.

One underlying cause of this phenomenon is the variation in intrinsic class difficulty. Easier classes–those that are well-separated in the feature space and show high clean accuracy–tend to preserve or improve their performance under adversarial training. In contrast, harder classes–with higher sample variability or overlap with others–often see a decline in robust accuracy. From a geometric standpoint, adversarial training tends to shift decision boundaries in ways that favor easier classes,

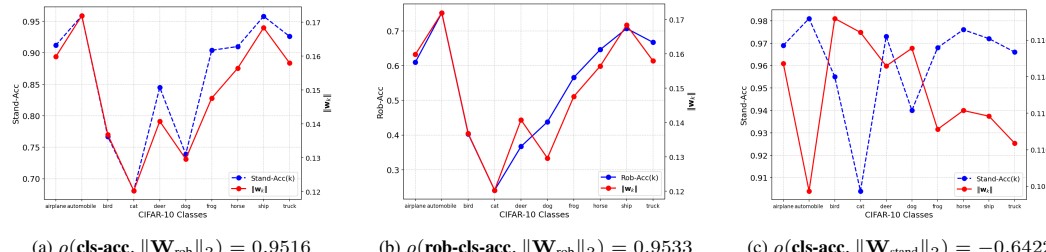

(a) $\rho(\mathbf{cls\text{-}acc}, \|\mathbf{W}_{\mathrm{rob}}\|_2) = 0.9516$     (b) $\rho(\mathbf{rob\text{-}cls\text{-}acc}, \|\mathbf{W}_{\mathrm{rob}}\|_2) = 0.9533$     (c) $\rho(\mathbf{cls\text{-}acc}, \|\mathbf{W}_{\mathrm{stand}}\|_2) = -0.6422$

Figure 1: Correlation ($\rho$) of Class-wise Standard and Robust Accuracies for adversarially trained model and standard trained model (non-adversarially robust trained model). cls-acc and rob-cls-acc are standard and robust class-wise accuracies against 20 step PGD, respectively. $\|\mathbf{W}_{1:C}^{\mathrm{stand}}\|_2$ and $\|\mathbf{W}_{1:C}^{\mathrm{rob}}\|_2$ are $\ell_2$ norms of class-wise head parameters for standard and robust trained model. We do not provide the $\rho(\mathbf{rob\text{-}cls\text{-}acc}, \|\mathbf{W}_{1:C}^{\mathrm{stand}}\|_2)$ for standard trained model since it has robust accuracies of 0% for all classes.

leaving harder ones more vulnerable to misclassification [20]. As a result, while average robustness improves, class-wise fairness deteriorates.

In this work, we take a new perspective on this issue by examining the role of the final classification layer (i.e., the head) in mediating class-wise disparities (Figure 1). We discover a strong correlation between the norms of the head parameters and class-wise robust accuracies. Specifically, adversarial training implicitly induces norm imbalances among head weights, which we find to be closely tied to class difficulty and robustness.

While adversarial training has shown promise in improving model robustness, it often introduces class-wise performance disparities, particularly affecting head parameters. Most previous approaches attempt to mitigate this by reweighting samples or remarging perturbation budgets, but they generally rely on hyperparameter tuning and validation sets, limiting their practical applicability. In contrast, our proposed HWNwB (Head Weights Normalization with Bias) and Deco-SAM methods are designed to directly address this imbalance without requiring extensive tuning, reducing both computational overhead and implementation complexity. By focusing on head weight norms, our approach effectively balances robustness across classes while maintaining overall model performance, representing a significant advancement over existing methods.

Our contributions are summarized as follows:

- We identify a strong correlation between the norms of class-specific head parameters and class-wise robust accuracies in adversarially trained models.

- We theoretically and empirically demonstrate that adversarial training induces imbalances in these norms, which contributes to performance disparities across classes.

- We propose lightweight algorithms that directly mitigate norm disparities at the head level through weight normalization or minimal post-training fine-tuning, without modifying the feature extractor or requiring a validation set.

- Our methods are compatible with a wide range of adversarial training algorithms (e.g., PGD-AT, TRADES, MART, ARoW) and incur negligible computational overhead.

- Through extensive experiments, we show that our approach significantly reduces class-wise disparities in both standard and robust accuracies while maintaining overall robustness.

Our findings provide a novel and practical approach to improving robust fairness, offering new insights into the structural origins of class-wise disparity in adversarial training and how it can be mitigated efficiently.

## 2 PRELIMINARIES

### 2.1 ROBUST POPULATION RISK

Let $\mathcal{X} \subset \mathbb{R}^d$ be an input space and a label set $\mathcal{Y} = \{1, \cdots, C\}$. Let $f : \mathcal{X} \to \mathbb{R}^C$ be a scoring function that produces a vector of predictive probabilities $\mathbf{p}(\boldsymbol{x}) = \mathrm{softmax}(f(\boldsymbol{x})) \in \mathbb{R}^C$ for each class. We define the classification function $h_f(\boldsymbol{x}) = \underset{k \in \mathcal{Y}}{\mathrm{argmax}} f_k(\boldsymbol{x}) \in \mathcal{Y}$, which assigns the input to

the class with the highest score. Additionally, let $\mathcal{B}_p(\boldsymbol{x}, \varepsilon) = \{\boldsymbol{x}' \in \mathcal{X} : \|\boldsymbol{x} - \boldsymbol{x}'\|_p \leq \varepsilon\}$ denote the $\varepsilon$-ball around $\boldsymbol{x}$ in the $p$-norm, and $\mathbb{1}(\cdot)$ be the indicator function. In the context of adversarial training, we aim to minimize the robust population risk, defined as:

$$\mathcal{R}_{\text{rob}}(f) = \mathbb{E}_{(\mathbf{X}, Y)} \max_{\mathbf{X}' \in \mathcal{B}_p(\mathbf{X}, \varepsilon)} \mathbb{1}\{Y \neq h_f(\mathbf{X}')\}. \tag{1}$$

This risk measures the worst-case expected misclassification rate within the $\varepsilon$-neighborhood of each input. If there exists an adversarial example $x' \in \mathcal{B}_p(\boldsymbol{x}, \varepsilon)$ that leads to the misclassification of $y$, the value of the 0-1 loss function is 1; otherwise, it is 0. The primary goal of adversarial training is to find an optimal scoring function $f$ (or equivalently, the predictive probability function $\mathbf{p}$) that minimizes this robust risk.

## 2.2 Algorithms for Adversarial Robustness

Recent works on defending against adversarial attacks, such as PGD-AT [12], TRADES [27] and ARoW [22], are grounded in minimizing theoretical bounds on the robust risk. PGD-AT directly minimizes the empirical risk, whereas TRADES and ARoW minimize the regularized empirical robust risk.

## 3 Related Works

The concept of robust fairness was first highlighted in the empirical survey by [3] and the theoretical study of [20]. Both studies observed that even when the training dataset contains an equal number of samples per class, there exists an inter-class discrepancy in terms of accuracy and robustness. [20] approached this issue theoretically by assuming that data follows a Gaussian mixture distribution with differing variances for each class, showing that adversarial training inevitably leads to this imbalance. [3] examined the potential of adapting long-tail techniques to address robust fairness in adversarial training.

Most existing solutions to this problem rely on class-wise weighting or regularization approaches [20; 13; 11; 18; 28], which are commonly used in long-tail learning techniques. Specifically, many algorithms for robust fairness employ class-wise weighting methods to adjust for inter-class imbalances [20; 11; 18; 28]. These approaches minimize the following loss function to address inter-class imbalances:

$$\frac{1}{n} \sum_{i=1}^{n} w_{\boldsymbol{\phi}}(\boldsymbol{x}_i, y_i) \ell_{\varepsilon}^{\text{rob}}(f_{\boldsymbol{\phi}}(\boldsymbol{x}_i), y_i), \tag{2}$$

where $n$ is the number of samples, $\phi$ is the parameters of $f$, $w_{\boldsymbol{\phi}}(\boldsymbol{x}_i, y_i)$ denotes the weight assigned to each sample, and $\ell_{\varepsilon}^{\text{rob}}$ is a surrogate risk used to approximate the robust risk (1). Common examples include PGD-AT [12], TRADES [27], ARoW [22], and MART [17]. Another approach, known as the remargin method [20], assigns different perturbation budgets $\varepsilon$ to each class, aiming to mitigate class-specific vulnerabilities. This is expressed as:

$$\frac{1}{n} \sum_{k=1}^{C} \sum_{i=1}^{n_k} w_{\boldsymbol{\phi}}(\boldsymbol{x}_i, y_i = k) \ell_{\varepsilon_k}^{\text{rob}}(f_{\boldsymbol{\phi}}(\boldsymbol{x}_i), y_i = k), \tag{3}$$

where $n_k$ is the number of samples assigned to class $k$, and $n = \sum_{k=1}^{C} n_k$. In addition to these weighting methods, long-tail techniques also include strategies for aligning decision boundaries across classes, which can further improve fairness. Building upon these approaches, in this paper, we conduct both theoretical and empirical analyses to uncover the optimization-driven causes of class disparity in adversarial training. This analysis guides the development of a novel and practical decision boundary alignment method, providing an effective solution to mitigate class-wise disparities in adversarial training.

## 4 Why Does the Disparity of Class-Wise Accuracies Occur?

In this section, we investigate why class-wise accuracy disparities arise in adversarial training compared to standard training in multi-class classification, drawing on both theoretical analysis and empirical observations. All corresponding proofs are provided in the Appendix.

## 4.1 THEORETICAL ANALYSIS

We start with a neural network composed of two parts - a feature extractor $\psi : \mathcal{X} \to \mathbb{R}^p$ and a head $h : \mathbb{R}^p \to \mathbb{R}^C$. Let $h$ be parameterized by weights $\mathbf{W}_{1:C} = (\mathbf{W}_1, \ldots, \mathbf{W}_C)$ and biases $\boldsymbol{b} = (\boldsymbol{b}_1, \ldots, \boldsymbol{b}_C)$, i.e., $h(\psi(\boldsymbol{x})) = (\mathbf{W}_k^\top \psi(\boldsymbol{x}) + \boldsymbol{b}_k)_{k=1}^C$. Let $p_k(\boldsymbol{x})$ be the $k$-th element of the prediction probability $\mathbf{p}(\boldsymbol{x})$, and $s_k(\boldsymbol{x}) = \mathbf{W}_k^\top \psi(\boldsymbol{x})$. Let $\boldsymbol{x}^{\text{adv}}$ be an adversarial example corresponding to a clean input $\boldsymbol{x}$, such that $\ell_{\text{ce}}(f(\boldsymbol{x}^{\text{adv}}), y) > \ell_{\text{ce}}(f(\boldsymbol{x}), y)$, or equivalently, $p_y(\boldsymbol{x}^{\text{adv}}) < p_y(\boldsymbol{x})$, where $p_y(\boldsymbol{x})$ denotes the predicted probability of the true class $y$. In addition, let $\boldsymbol{\theta}_{\psi(\boldsymbol{x}),k}$ denote the angle between the feature representation $\psi(\boldsymbol{x})$ and the weight vector $\mathbf{W}_k$ of class $k$.

To examine the type of bias present in the final model resulting from adversarial training, we focus on the case where the training loss has been sufficiently minimized. Under this assumption, it is reasonable to approximate $\psi(\boldsymbol{x}^{\text{adv}}) \simeq \psi(\boldsymbol{x})$, since prior work has shown that adversarial training tends to better preserve robust feature representations compared to standard training [24; 26]. Then, both $\cos(\boldsymbol{\theta}_{\psi(\boldsymbol{x}),y})$ and $\cos(\boldsymbol{\theta}_{\psi(\boldsymbol{x}^{\text{adv}}),y})$ are expected to be large; refer to Proposition 2 in Section A.2 for a more rigorous formulation.

Before presenting the main theorem, we define the following two relative measures.

**Definition 1.** *We define the class-specific gradient gap measure and its expected version as*

$$\delta(\boldsymbol{x}, y) := \left| \frac{\partial \ell_{ce}(f(\boldsymbol{x}^{adv}), y)}{\partial \|\mathbf{W}_y\|_2} \right| - \left| \frac{\partial \ell_{ce}(f(\boldsymbol{x}), y)}{\partial \|\mathbf{W}_y\|_2} \right|, \Delta_k := \mathbb{E}_{(\mathbf{X}, Y=k)} \delta(\mathbf{X}, Y), \tag{4}$$

*respectively.*

**Definition 2.** *Define the* hardness *of class $k$ by $H_k := \mathbb{E}_{(\mathbf{X}, Y=k)}[p_k(\mathbf{X}) - p_k(\mathbf{X}^{adv})]$. A class $c_{hard}$ is said to be* harder *than a class $c_{easy}$ iff $H_{c_{hard}} > H_{c_{easy}}$.*

***Remark* 1.** The scalar gap $\delta(\boldsymbol{x}, y)$ measures, for each sample, how much the adversarial example amplifies the gradient magnitude with respect to the $y$-th head-norm compared with the clean sample; a larger $\delta$ therefore reflects a stronger push that drives $\|\mathbf{W}_y\|_2$ upward during SGD. Meanwhile, the hardness index $H_k$ is the class-level average drop in the correct-class posterior $p_k$ induced by the adversarial attack, so a larger $H_k$ indicates that class $k$ is inherently more vulnerable (i.e., harder) against adversarial attack.

**Proposition 1.** $\Delta_k = \mu_Z H_k$ *holds. Consequently, if a class $c_{hard}$ is harder than class $c_{easy}$ ($H_{c_{hard}} > H_{c_{easy}}$), then $\Delta_{c_{hard}} > \Delta_{c_{easy}}$.*

***Remark* 2.** $H_k$ is the *average drop in class probability* caused by the adversarial attack. Proposition 1 shows that this drop translates linearly into the gradient gap $\Delta_k$, so that harder classes necessarily incur larger $\Delta_k$. This observation directly justifies the assumption $\Delta_h > \Delta_e$, which is a key condition underlying the drift dynamics analyzed in the Theorem 1 below.

**Theorem 1.** *Run stochastic gradient descent with learning rate $\eta$ for $T$ iterations using the adversarial loss. Let $\Delta_k := \mathbb{E}_{(\mathbf{X}, Y=k)} \delta(\mathbf{X}, Y)$ be the class-specific expected gradient gap. Then,*

$$\mathbb{E}\|\mathbf{W}_k^{(T)}\|_2 = \|\mathbf{W}_k^{(0)}\|_2 + \eta T \Delta_k. \tag{5}$$

*Consequently, if a class $c_{hard}$ is* harder *than a class $c_{easy}$ ($\Delta_{c_{hard}} > \Delta_{c_{easy}}$), there exists $T^*$ such that $\mathbb{E}\|\mathbf{W}_{c_{hard}}^{(T)}\|_2 > \mathbb{E}\|\mathbf{W}_{c_{easy}}^{(T)}\|_2$ for all $T \geq T^*$.*

These results demonstrate that a larger gradient gap $\Delta_k$ drives a steady increase in the $\ell_2$-norm of the corresponding class head. As training proceeds, the norms of harder classes grow more rapidly than those of easier ones, thereby widening robustness disparities. Theorem 1 formally characterizes this gradient-imbalance effect, shedding light on why adversarial training can exacerbate class-wise robustness differences and motivating the normalization strategies proposed in Section 5.

## 4.2 EMPIRICAL OBSERVATIONS

In addition to the previous theoretical correlation between $\mathbf{W}_{1:C}$ and class-wise robust accuracies, we also provide empirical observations on them. Specifically, we observe the correlation between $\mathbf{W}_{1:C}$ and **rob-acc**, where $\|\mathbf{W}_{1:C}\|_2 := (\|\mathbf{W}_1\|_2, \|\mathbf{W}_2\|_2, \ldots, \|\mathbf{W}_C\|_2)$ and **rob-acc** $:= (\text{rob-acc}(1), \text{rob-acc}(2), \ldots, \text{rob-acc}(C))$, where $\text{rob-acc}(c)$ represents the robust accuracy of the

$c$-th class. Figures 1a and 1b show that the class-wise standard and robust accuracies of the adversarially trained model are highly correlated with the norms of the head parameters. Conversely, Figure 1c reveals that in the standard trained model, there is no significant correlation between them. To compare the disparity in head parameter norms between adversarially robust and standard models, we use the ratio $\max_k \|\mathbf{W}_k\|_2 / \min_k \|\mathbf{W}_k\|_2$. This index measures the relative disparity in the norms of head weights, with a minimum value of 1, indicating that all norms are equal when the index is exactly 1. The value for the standard trained model is 1.08, whereas for the adversarially robust trained model, it is 1.43, highlighting a significant difference in norm disparity in adversarially trained models. These observations suggest that adversarial training algorithms inherently induce bias in the head parameters, causing the norms of more challenging classes to increase while those of relatively easier classes remain smaller, which coincides with the theoretical analysis from Section 4.1. An increase in norm magnitude implies an expansion of the decision boundary region for the corresponding class in multi-class classification problems. We identify this phenomenon as **implicit bias** in adversarially robust training.

## 5 PROPOSED METHODS

In the previous section, we identified that class-wise robustness disparities primarily arise from the implicit bias introduced during adversarial training, particularly affecting the head parameters. To address this issue, we propose two complementary methods that specifically target this imbalance, focusing on efficiency without the need for extensive hyperparameter tuning or additional validation sets.

### 5.1 HEAD WEIGHTS NORMALIZATION

To directly address the gradient accumulation imbalance highlighted in Section 4, we introduce the Head Weights Normalization with Bias (HWNwB) method. Unlike traditional weight normalization approaches, HWNwB aims to stabilize head norms without relying on separate validation sets, and effectively reduces the class-wise robustness gap while maintaining computational efficiency. This approach mitigates the overfitting risk for challenging classes with large gradient updates, promoting balanced head norms across all classes. Formally, for a fixed feature extractor and bias terms, the head weights are normalized as $\widetilde{\mathbf{W}}_k := \frac{\mathbf{W}_k}{\|\mathbf{W}_k\|_2}$. Then, the label prediction is given as $\underset{k \in \mathcal{Y}}{\arg\max}(\widetilde{\mathbf{W}}_k^\top \psi(\boldsymbol{x}) + b_k)$ with normalized weights $\widetilde{\mathbf{W}} = (\widetilde{\mathbf{W}}_1, \cdots, \widetilde{\mathbf{W}}_C)$.

**Effect of Bias Terms in Normalized Weight Models**  In this paragraph, we examine the impact of bias terms on confusing classes in a normalized weight setting. Consider the score for class $c$, defined as $\mathbf{W}_c^\top \psi(\boldsymbol{x}) + b_c$. The decision boundary between classes 1 and 2 is given by $\boldsymbol{x} : \mathbf{W}_1^\top \psi(\boldsymbol{x}) + b_1 = \mathbf{W}_2^\top \psi(\boldsymbol{x}) + b_2$, with its distance from the origin expressed as $\frac{|b_1 - b_2|}{\|\mathbf{W}_1 - \mathbf{W}_2\|_2}$.

When weight vectors are normalized (i.e., $\|\mathbf{W}_1\|_2 = \|\mathbf{W}_2\|_2 = 1$), this distance becomes $\frac{|b_1 - b_2|}{\sqrt{2(1 - \mathbf{W}_1^\top \mathbf{W}_2)}}$. For similar classes, where $\mathbf{W}_1^\top \mathbf{W}_2$ is high, the bias terms $b_1$ and $b_2$ have a critical influence on the positioning of the boundary. Figures 2b and 2c show that the correlation of heads for confusing classes, such as *cat* and *dog*, is the highest.

Furthermore, Figure 3 illustrates that small variations in the bias term for similar (confusing) classes cause more sensitive shifts in the decision boundary compared to distinct classes, significantly impacting class separation. We present additional experiments in Section E.2 that further emphasize the importance of bias terms in such settings.

### 5.2 DECOUPLED SAM

Our approach builds on the idea that a high head norm correlates with a wider decision boundary [8], signifying an easier class that needs only minor adjustments. On the other hand, a low head norm corresponds to a narrower decision boundary [8], indicating a more difficult class that requires larger modifications. This understanding allows us to better balance the model's robustness across

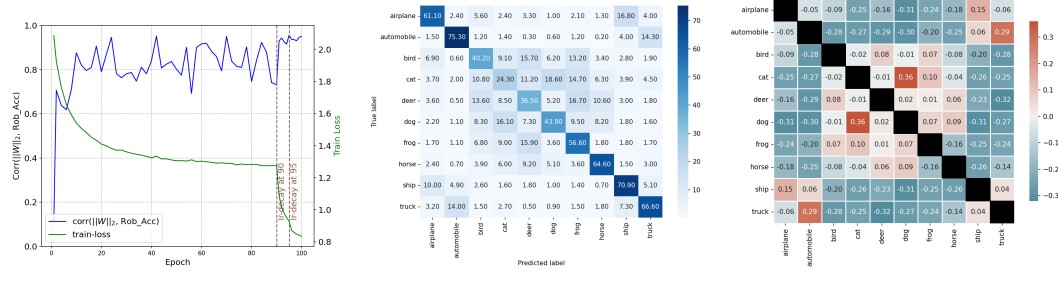

(a) Training Dynamics of $\mathbf{corr}(\|\mathbf{W}_{1:C}^{\mathbf{rob}}\|_2, \mathbf{rob\text{-}acc})$

(b) Confusion Matrix of Rob-Acc (PGD-20).

(c) Correlation Matrix of Weights of Head

Figure 2: Figures illustrate the training dynamics and final epoch models of PGD-AT without a validation set. To prevent robust overfitting, a learning rate decay is applied just before the 5th and 10th epochs of the total training epochs, as done in [15], in order to select models that have not overfitted.

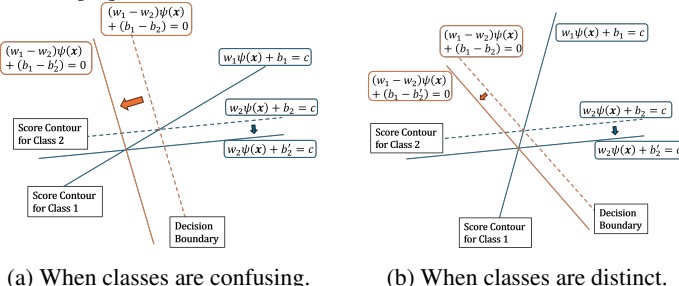

(a) When classes are confusing.      (b) When classes are distinct.

Figure 3: An illustration on decision boundary shifts, given bias term shifts. Score contours are calculated via arbitrary level $c$. Higher weight similarity leads to larger decision boundary shifts.

different classes. This concept of head norms and decision boundaries is central to the functionality of DecoSAM, enabling more effective class-wise adjustments.

Our approach is motivated by the Sharpness Aware Minimization (SAM) [6] optimizer, which optimizes:

$$\min_{\boldsymbol{\theta}} \frac{1}{n} \sum_{i=1}^{n} \max_{\|\delta\|_2 \leq \rho} \ell_{\mathrm{ce}}(f_{\boldsymbol{\theta}+\boldsymbol{\delta}}(\boldsymbol{x}_i), y_i) \tag{6}$$

where $\ell_{\mathrm{ce}}$ is the cross-entropy loss. Geometrically, it finds a flatter minimizer than standard training.

Modifying SAM and motivated by HWNwB, DecoSAM minimizes the following empirical risk:

$$\min_{\mathbf{W}} \frac{1}{n} \sum_{k=1}^{C} \sum_{i=1}^{n_k} \max_{\|\boldsymbol{\delta}_k\|_2 \leq \rho_k} \ell_{\mathrm{ce}}(h_{\mathbf{W}_k+\boldsymbol{\delta}_k} \circ \psi(\boldsymbol{x}_i), y_i) \tag{7}$$

where $n = \sum_{k=1}^{C} n_k$. In this optimization, it is important to select $\rho_k$. Inspired by HWNwB, we select it as

$$\nu_k = \frac{\exp(-\tau\|\mathbf{W}_k\|_2)}{\sum\limits_{k=1}^{C} \exp(-\tau\|\mathbf{W}_k\|_2)} \tag{8}$$

with constrained $\sum_{k=1}^{C} \nu_k = \rho$.

Our approach leverages the relationship between head norm and decision boundary width to balance model robustness across classes, allowing us to adaptively adjust the model's behavior for different classes. Specifically, for difficult classes with low head norms and narrow decision boundaries, we make larger adjustments to improve the robustness of the worst-performing class, while for easier classes with high head norms and wide decision boundaries, we make smaller adjustments to maintain robustness for that class. DecoSAM employs this adaptive strategy to align head weights with the fixed feature extractor, enhancing robust fairness through class-specific adjustments based on head norms. Additionally, SAM's tendency to locate flat minima in the loss landscape further boosts DecoSAM's robust fairness.

This dual approach, which combines class-specific adaptations with SAM's inherent robustness properties, yields a model that is well calibrated and robust for varying class difficulties. To implement this, we replace the standard cross-entropy loss ($\ell_{ce}$) with surrogate loss functions for robust risk, utilizing methods such as PGD-AT [12], TRADES [27], MART [17], and ARoW [22]. The complete procedure is summarized in Algorithm 1.

## 5.3 COMPARISON WITH EXISTING ALGORITHMS

**Comparison with FRL [20]**  Theoretical analysis in Fair Robust Learning (FRL) [20] reveals that adversarial training expands the decision boundary for easier classes while compressing it for harder ones. However, FRL focuses on binary classification and does not provide an optimization-based explanation for the observed class-wise disparity. In contrast, our work addresses the **multi-class** setting and identifies **gradient imbalance relative to head weight norms** as the core reason for disparity. Moreover, FRL is tied to TRADES [27] and lacks algorithm-agnostic flexibility. Our proposed methods are **algorithm-agnostic** and compatible with various adversarial training frameworks, including PGD-AT [12] and MART [17]. We further contribute optimization-based insights and introduce simple, practical strategies to mitigate class-wise unfairness.

**Comparison with Existing Algorithms**  Most robust fairness methods mitigate class-wise disparities through re-weighting or class-specific budgets, typically modifying the loss function. In contrast, our work uniquely applies normalization techniques to address class-wise robustness gaps under class-balanced data. First, unlike $\tau$-normalization [7], which removes bias terms post-training, we show that preserving them is crucial for fairness and robust accuracy. Second, while cRT helps in long-tailed setups, it fails under adversarial training on balanced data–unlike our HWNwB and Deco-SAM. Lastly, our methods are computationally efficient: HWNwB applies simple weight normalization, and Deco-SAM adds only one SAM epoch, yet both achieve strong robust fairness.

Our approach is grounded in the insight that head weight norms correlate with class difficulty: larger norms indicate easier classes with wider margins. By aligning weight directions while preserving critical bias terms, our methods efficiently promote fairness without compromising overall robustness.

Table 1: **Summary of Existing Works and Comparison to Proposed Methods.** $w_{\theta}(\boldsymbol{x}, y)$ and $\varepsilon_y$ represent class-wise re-weighting (CW-RW) and re-margin(RM) methods in (3), respectively. **Val.** indicates the necessity of a validation set. **Extensibility** is a boolean value that indicates whether a method can be applied to adversarial training algorithms such as PGD-AT [12], TRADES [27], MART [17], and ARoW [22].

| Method | $w_{\theta}(\boldsymbol{x}, y)$ | $\varepsilon_y$ | Val. | Extensibility | Remark |
|---|---|---|---|---|---|
| FRL [20] | ✓ | ✓ | ✓ | ✗ | First Work |
| FAT [13] | ✗ | ✗ | ✗ | ✓ | Variance Regularization |
| WAT [11] | ✓ | ✗ | ✓ | ✗ | CW-RW Loss |
| CFA [18] | ✓ | ✓ | ✓ | ✗ | CW-RW loss, CW-RM loss , Weight Averaging |
| FAAL [28] | ✓ | ✗ | ✗ | ✓ | Distributionally Robust Optimization |
| HWNwB | ✗ | ✗ | ✗ | ✓ | Aligning Decision Boundary |
| DecoSAM | ✗ | ✗ | ✗ | ✓ | Aligning Decision Boundary |

## 6 EXPERIMENTS

We utilize CIFAR-10 [10] and CIFAR-100 [10], which are widely recognized benchmark datasets for evaluating robust fairness in prior studies [20; 11; 18; 28]. Additionally, we incorporate STL-10 [4] and OfficeHome [16] to assess the effectiveness of our approach in higher-resolution settings. The results for CIFAR-100, OfficeHome and ImageNet100 are presented in the Appendix. To evaluate the effect of our algorithm across different levels of model capacity, we conduct experiments using three variants of WideResNet—WRN-28-2, WRN-28-5, and WRN-28-10 [25], as well as additional architectures including ResNet-50 and a Vision Transformer (ViT) model to further assess generality across both convolutional and transformer-based networks in the Appendix. Our aim is to validate the generality of our method by applying it to a range of adversarial training algorithms that minimize surrogate robust risk objectives, including PGD-AT [12], TRADES [27], MART [17], and ARoW [22]. While ARoW is not explicitly designed to enhance robust fairness, it has been reported to exhibit fairness-improving properties [22]. In our experiments, we also include a comparison with FAAL and evaluate the combination ARoW + DecoSAM, highlighting the improvements in both average and worst-class accuracies. Note that we do not include the full ImageNet-1k dataset in our evaluation, as its large number of classes makes robust fairness assessment difficult—worst-class

Table 2: **Comparison of HWNwB and DecoSAM Performance on Baseline Algorithms on CIFAR-10. PGD** and **AA** indicate the robust accuracy under a 20-step PGD attack and the AutoAttack, respectively. **WC** indicates the worst-class robust accuracy, **STD** indicates the standard deviation of class-wise robust accuracies, and **Max-Min** indicates the difference between the highest and lowest class-wise robust accuracies.

| Method | Clean(↑) | PGD(↑) | WC(↑) | STD(↓) | Max-Min(↓) | corr(‖W‖₂, PGD) | AA(↑) | WC(↑) | STD(↓) | Max-Min(↓) |
|---|---|---|---|---|---|---|---|---|---|---|
| **CIFAR-10 (WRN-28-2)** | | | | | | | | | | |
| PGD-AT | **80.85** | 49.20 | 19.70 | 17.47 | 53.70 | 0.9314 | **45.27** | 13.70 | 19.13 | 58.00 |
| + HWNwB | 79.47 | 52.74 | 29.10 | **13.44** | 41.30 | – | 43.50 | 18.60 | **14.71** | 46.10 |
| + DecoSAM | 79.69 | **52.79** | **31.50** | 13.64 | **39.97** | – | 44.26 | **21.90** | 15.01 | **44.33** |
| TRADES | **78.98** | 49.30 | 22.40 | 16.83 | 48.10 | 0.8964 | **45.33** | 17.20 | 18.15 | 51.30 |
| + HWNwB | 78.74 | **51.42** | 29.90 | **14.27** | 41.70 | – | 44.96 | 19.40 | **15.72** | 48.60 |
| + DecoSAM | 78.21 | 51.30 | **30.57** | 15.59 | **40.87** | – | 45.00 | **22.53** | 16.23 | **43.97** |
| MART | **77.13** | 51.44 | 20.90 | 17.59 | 52.10 | 0.9129 | **46.27** | 12.30 | 20.40 | 57.90 |
| + HWNwB | 75.12 | 52.75 | 26.90 | 15.29 | 45.80 | – | 44.03 | 15.80 | 16.11 | 50.00 |
| + DecoSAM | 75.75 | **53.12** | **27.47** | **14.96** | **45.27** | – | 44.45 | **19.90** | **16.07** | **45.97** |
| ARoW | 79.82 | 50.05 | 23.90 | 15.72 | 45.50 | 0.9437 | **45.97** | 18.70 | 17.01 | 48.70 |
| + HWNwB | 78.61 | 51.88 | **36.90** | **12.44** | **33.70** | – | 44.22 | 28.10 | 14.10 | 36.50 |
| + DecoSAM | 78.73 | **52.34** | 35.43 | 12.88 | 35.33 | – | 44.85 | **28.60** | **13.65** | **36.17** |
| **CIFAR-10 (WRN-28-5)** | | | | | | | | | | |
| PGD-AT | **86.00** | 53.94 | 24.20 | 16.07 | 51.30 | 0.9515 | **49.50** | 17.60 | 18.03 | 55.70 |
| + HWNwB | 85.09 | **56.68** | **39.10** | 12.31 | **34.80** | – | 48.25 | 29.10 | **13.83** | **38.90** |
| + DecoSAM | 85.52 | 56.55 | 38.93 | **12.22** | 35.37 | – | 49.09 | **30.70** | 14.60 | 39.93 |
| TRADES | 83.52 | 53.85 | 29.60 | 15.68 | 46.50 | 0.8967 | 50.65 | 23.90 | 17.17 | 51.10 |
| + HWNwB | 82.93 | **56.13** | **37.60** | 13.54 | 39.00 | – | 49.88 | 26.20 | **14.77** | 46.10 |
| + DecoSAM | 83.01 | 56.05 | 36.00 | 14.84 | **38.23** | – | 50.24 | **29.17** | 15.92 | **41.27** |
| MART | **82.66** | 55.00 | 25.80 | 15.92 | 52.00 | 0.9528 | **49.77** | 17.60 | 18.78 | 58.00 |
| + HWNwB | 80.28 | **56.88** | **36.50** | **13.44** | **37.80** | – | 48.26 | 24.60 | 16.12 | 44.60 |
| + DecoSAM | 80.66 | 56.75 | 33.97 | 14.06 | 40.60 | – | 48.90 | **25.30** | 15.50 | 43.90 |
| ARoW | **84.18** | 53.46 | 27.10 | 15.26 | 48.50 | 0.9328 | 50.36 | 22.70 | 16.42 | 51.40 |
| + HWNwB | 83.43 | 56.21 | **43.70** | **11.87** | **30.30** | – | **48.36** | 30.05 | **13.08** | 36.90 |
| + DecoSAM | 82.82 | **56.45** | 37.57 | 13.08 | 35.37 | – | 49.29 | **31.30** | 13.89 | **36.43** |
| **CIFAR-10 (WRN-28-10)** | | | | | | | | | | |
| PGD-AT | **87.74** | 52.75 | 22.50 | 16.18 | 51.10 | 0.9061 | **50.06** | 19.70 | 17.29 | 53.50 |
| + HWNwB | 87.30 | **56.89** | **42.10** | **11.23** | **31.20** | – | 49.35 | **30.70** | **13.98** | **39.90** |
| + DecoSAM | 87.36 | 56.31 | 39.53 | 12.16 | 34.57 | – | 49.64 | 29.87 | 14.63 | 42.43 |
| TRADES | 85.35 | 55.71 | 29.00 | 15.57 | 48.00 | 0.8741 | 52.86 | 25.30 | 16.89 | 50.80 |
| + HWNwB | 84.90 | 57.77 | **39.00** | **13.37** | 37.00 | – | 52.21 | 30.40 | **14.69** | 42.30 |
| + DecoSAM | 84.25 | **57.78** | 38.03 | 14.53 | 37.77 | – | 52.35 | **31.10** | 15.71 | **42.23** |
| MART | **85.30** | 56.64 | 31.60 | 14.95 | 44.90 | 0.9460 | **51.32** | 22.00 | 17.84 | 52.70 |
| + HWNwB | 83.64 | **58.82** | **39.30** | **12.83** | **35.80** | – | 49.99 | 30.00 | **14.53** | **41.40** |
| + DecoSAM | 84.05 | 58.68 | 38.03 | 12.95 | 38.23 | – | 50.64 | **31.63** | 14.79 | 41.70 |
| ARoW | **85.97** | 55.23 | 27.30 | 15.75 | 49.30 | 0.8910 | 52.27 | 23.00 | 16.73 | 52.40 |
| + HWNwB | 84.90 | **57.87** | **40.90** | 13.58 | **36.70** | – | **50.42** | 31.50 | 14.68 | **36.30** |
| + DecoSAM | 84.66 | 57.82 | 38.90 | **13.22** | 37.47 | – | 50.82 | **32.90** | **14.01** | 38.47 |

Table 3: **Comparison of HWNwB and DecoSAM Performance on Baseline Algorithms on STL-10. PGD** and **AA** indicates the robust accuracy under a 20-step PGD attack and the AutoAttack. **WC** indicates the worst-class robust accuracy, **STD** indicates the standard deviation of class-wise robust accuracies, and **Max-Min** indicates the difference between the highest and lowest class-wise robust accuracies.

| Method | Clean(↑) | PGD(↑) | WC(↑) | STD(↓) | Max-Min(↓) | corr(‖W‖₂, PGD) | AA(↑) | WC(↑) | STD(↓) | Max-Min(↓) |
|---|---|---|---|---|---|---|---|---|---|---|
| **STL-10 (WRN-28-5)** | | | | | | | | | | |
| PGD-AT | **81.28** | 65.51 | 34.75 | 18.40 | 54.38 | 0.6502 | **62.40** | 26.00 | 20.48 | 62.12 |
| + HWNwB | 79.57 | **67.10** | 41.75 | **17.23** | 47.38 | – | 61.58 | 34.00 | **18.13** | **49.50** |
| + DecoSAM | 79.92 | 66.71 | **42.63** | 17.47 | **46.75** | – | 61.59 | **36.29** | 19.42 | 50.33 |
| TRADES | **79.67** | 62.41 | 34.00 | 18.13 | 51.50 | 0.5379 | **58.69** | 24.62 | 20.44 | 59.12 |
| + HWNwB | 78.54 | **64.06** | 40.12 | **17.36** | 46.75 | – | 58.10 | 31.62 | 20.51 | 50.75 |
| + DecoSAM | 78.43 | 63.62 | 39.83 | 17.85 | 46.50 | – | 58.03 | **34.54** | **19.88** | **48.62** |
| ARoW | **80.65** | 63.95 | 34.88 | 17.78 | 51.12 | 0.7113 | **60.44** | 25.62 | 19.90 | 58.75 |
| + HWNwB | 80.05 | **65.74** | 43.50 | **16.90** | 44.12 | – | 59.15 | 34.88 | 19.70 | 49.88 |
| + DecoSAM | 80.22 | 65.33 | **43.87** | 17.31 | **42.55** | – | 59.58 | **35.71** | 19.57 | 47.71 |

accuracy typically collapses to near-zero in such high-class settings. Instead, we report experiments on ImageNet-100, an extracted subset of ImageNet-1k that is widely used in robust robustness and fairness studies. This reduced-class version enables meaningful evaluation of class-wise disparity. The ImageNet-100 results, together with $\varepsilon$-sweep experiments, are included in the Appendix. Ablation studies on the effects of robust regularization intensity and bias terms in the proposed algorithms and $\tau$ and $rho$ in eqn (7) are provided in the Appendix.

**Training Setups**  We follow the experimental setting of Pang et al. [15] and select the model from the last epoch to avoid using a validation set. To generate adversarial examples, we employ a 10-step PGD attack with a perturbation budget of $\varepsilon = 8/255$ and a step size of $2/255$. For the pretrained model, we use a weight decay of $5e^{-4}$, train for a total of 100 epochs, and employ a multi-step learning rate scheduler with learning-rate decays at epochs 90 and 95 for all algorithms. The regularization parameters for TRADES, MART, and ARoW are set to 6, 3, and 7, respectively. HWNwB applies simple head weight normalization without additional training, keeping the bias terms unchanged. DecoSAM with $\tau = 1$ in (7), on the other hand, performs training for only **one epoch** with fixed learning rate.

**Evaluation Setups**  To evaluate robust fairness, we adopt two complementary methods: a 20-step PGD attack using the same configuration as during training (perturbation budget $\varepsilon = 8/255$, step size $\alpha = 2/255$), and AutoAttack (AA) [5], a standardized ensemble of attacks known for providing reliable robustness evaluations. AA is particularly valuable in mitigating the effects of gradient obfuscation [1], where misleading gradients can result in overestimated robustness under weaker attacks. For comprehensive assessment, we report five key metrics: clean accuracy, robust accuracy, worst-class accuracy, standard deviation of class-wise accuracies, and the accuracy gap between the best and worst-performing classes. Among these, worst-class accuracy is the most critical metric, as it reflects the robustness of the most vulnerable class and serves as a widely adopted indicator of robust fairness in prior works [11; 20; 28; 18].

## 6.1 Performance Evaluation

Table 2 presents the performance of HWNwB and DecoSAM across various adversarially robust training algorithms–PGD-AT, TRADES, MART, and ARoW–on architectures of varying complexity. Overall, all algorithms exhibited a high correlation between weight norms and PGD robustness across architectures, with TRADES showing a slightly lower correlation. In terms of PGD accuracy, HWNwB generally outperformed DecoSAM, likely because weight normalization tends to equalize class scores more uniformly [19]. Despite this, DecoSAM achieved significant improvements in robust fairness under AutoAttack (AA), performing particularly well in worst-class robust accuracy and showing notable gains in overall robust accuracy under AA, thus proving its effectiveness in enhancing robust fairness across classes. Table 3 shows the performance of our algorithm. Similar to CIFAR-10, the correlation is high, demonstrating that our approach significantly improves robustness fairness.

**Combination with Existing Works**  In this paragraph, we integrate our methods with existing algorithms such as FRL [20], FAT [13], WAT [11], CFA [18], and FAAL [28]. Each algorithm is implemented using the default settings from the corresponding official repository, based on the WRN-28-5 architecture. The trained models are saved and then HWNwB and DecoSAM are applied, after which they are evaluated to assess their performance. Table 4 presents the results of combining HWNwB and DecoSAM with existing algorithms. We also observe that existing robust fairness algorithms induce a high correlation between head parameters and class-wise robust accuracies. Our methods demonstrate improvements in worst-class accuracy across all methods except FAAL. In FAAL, although worst-class accuracy decreases, overall accuracy increases.

**Effect of Bias Term in HWNwB and DecoSAM**  We conduct experiments with HWNwB and DecoSAM to examine the importance of bias terms in enhancing performance, particularly for the worst-performing classes. Our proposed algorithms, HWN w/ Bias and DecoSAM w/ frozen Bias, preserve the bias term to explore its impact, while HWN w/o Bias follows the traditional $\tau$-normalization technique by removing the bias term, and DecoSAM w/o frozen Bias allows the bias term to be updated alongside other parameters. As shown in Table 5 and discussed in Section 5.1, retaining the bias term significantly improves worst-class performance in both HWN and DecoSAM, underscoring its critical role in achieving robust fairness.

## 7 Conclusion and Future Work

This paper introduces a novel approach to improving robust fairness in adversarial training by uncovering a strong correlation between classifier head parameter norms and class-wise robust accuracies.

Table 4: **Combination with Existing Algorithms.**

| Method | CIFAR-10 (WRN-28-5) | | | | | |
|---|---|---|---|---|---|---|
| | **Clean(↑)** | **AA(↑)** | **WC(↑)** | **STD(↓)** | **Max-Min(↓)** | **corr($\|W\|_2$, PGD)** |
| FRL | **84.09** | **46.85** | 27.10 | 14.23 | **40.10** | 0.9056 |
| + HWNwB | 81.31 | 44.38 | 29.50 | **12.90** | 43.30 | – |
| + DecoSAM | 83.51 | 46.03 | **30.11** | 14.07 | 40.19 | – |
| FAT | **83.22** | **50.02** | 20.70 | 17.47 | 56.20 | 0.9469 |
| + HWNwB | 82.95 | 49.11 | 27.90 | **15.14** | 47.00 | – |
| + DecoSAM | 82.31 | 48.88 | **29.90** | 15.32 | **43.80** | – |
| CFA | 85.42 | **50.42** | 23.70 | 16.52 | 50.00 | 0.9075 |
| + HWNwB | 85.41 | 49.73 | 27.90 | 15.13 | 44.30 | – |
| + DecoSAM | **85.52** | 50.33 | **29.11** | **14.59** | **42.75** | – |
| FAAL | 81.19 | 48.81 | 31.70 | 11.86 | 33.90 | 0.9114 |
| + HWNwB | 77.32 | 44.91 | 31.90 | **11.12** | 32.90 | – |
| + DecoSAM | **81.96** | **50.11** | **32.10** | 11.37 | **32.80** | – |
| WAT | **83.62** | **50.50** | 21.50 | 16.90 | 53.00 | 0.9508 |
| + HWNwB | 83.13 | 49.93 | 30.30 | **14.74** | 41.40 | – |
| + DecoSAM | 83.19 | 50.22 | **30.60** | 15.34 | **39.80** | – |

Table 5: **Effect of Bias Terms of HWNwB and DecoSAM.**

| Method | CIFAR-10 (WRN-28-5) | | | | |
|---|---|---|---|---|---|
| | **Clean(↑)** | **AA(↑)** | **WC(↑)** | **STD(↓)** | **Max-Min(↓)** |
| PGD-AT | **86.00** | **49.50** | 17.60 | 18.03 | 55.70 |
| + HWN w/ Bias | 85.09 | 48.25 | 29.10 | **13.83** | **38.90** |
| + HWN w/o Bias | 86.23 | 48.14 | 23.10 | 16.50 | 48.20 |
| + DecoSAM w/ frozen Bias | 85.52 | 49.09 | **30.70** | 14.60 | 39.93 |
| + DecoSAM w/o frozen Bias | 85.28 | 48.97 | 28.60 | 16.00 | 42.70 |

We show that adversarial training induces imbalances in head norms, which in turn lead to disparities in class-wise performance. To mitigate this issue, we propose an algorithm that fine-tunes head parameters without requiring a validation set or modifying the feature extractor, effectively reducing accuracy gaps across classes while preserving overall robustness. Extensive experiments demonstrate that our method improves both fairness and robustness.

Despite these contributions, our analysis primarily focuses on linear classifier heads. Although many modern architectures employ non-linear heads such as MLPs, our theoretical results still apply because the final prediction is ultimately computed through a linear transformation in the last layer. By treating the input to this layer as the feature representation, our methods and theoretical insights remain valid. Nevertheless, a limitation of our current framework is that it mainly addresses robustness disparities at the classifier head level. Implicit biases can also emerge in the feature representation space, particularly under adversarial training, which tends to amplify such biases at both the feature and classifier levels.

As future work, we plan to extend our robustness-balancing framework beyond the classifier head to jointly address biases in both the feature and weight spaces. Such an extension would provide a more comprehensive and principled strategy for improving class-wise fairness and robustness against adversarial attacks.

**Reproducibility Statement** We have taken considerable care to guarantee the reproducibility of our findings in this study. For the theoretical results, we include full proofs in the Appendix. The source code for implementing our proposed model are provided in the supplementary material. Detailed information for the hyperparameters, datasets and experimental setup are given in Section D of Appendix.

**Use of Large Language Models** In the preparation of this manuscript, a large language model was utilized as a writing aid. Its role was strictly limited to improving grammar, rephrasing for clarity, and correcting typographical errors. The LLM did not contribute to the core research ideas, experimental design, or the analysis of results presented in this paper.

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

# APPENDIX

## A   THEORETICAL RESULTS

In this section, we provide detailed assumptions and proofs of the theoretical results.

**Gradient Formulas.**   We first consider the cross-entropy loss $\ell_{ce}$ for a multi-class classification task. The gradient with respect to the class score $s_k(\boldsymbol{x})$ is

$$\frac{\partial \ell_{ce}(f(\boldsymbol{x}), y)}{\partial s_k(\boldsymbol{x})} = p_k(\boldsymbol{x}) - y_k, \tag{9}$$

where $s_k(\boldsymbol{x})$ is the logit for class $k$:

$$s_k(\boldsymbol{x}) = \mathbf{W}_k^\top \psi(\boldsymbol{x}) = \|\mathbf{W}_k\|_2 \|\psi(\boldsymbol{x})\|_2 \cos(\boldsymbol{\theta}_{\psi(\boldsymbol{x}),k}), \tag{10}$$

and $\boldsymbol{\theta}_{\psi(\boldsymbol{x}),k}$ is the angle between $\mathbf{W}_k$ and the feature vector $\psi(\boldsymbol{x})$.

Since our goal is to investigate the effect of the head parameter norm on the loss, we compute the derivative of the loss with respect to $\|\mathbf{W}_k\|_2$:

$$\begin{aligned}
\frac{\partial \ell_{ce}(f(\boldsymbol{x}), y)}{\partial \|\mathbf{W}_k\|_2} &= \frac{\partial \ell_{ce}(f(\boldsymbol{x}), y)}{\partial s_k(\boldsymbol{x})} \cdot \frac{\partial s_k(\boldsymbol{x})}{\partial \|\mathbf{W}_k\|_2} \\
&= \begin{cases} (p_k(\boldsymbol{x}) - 1) \|\psi(\boldsymbol{x})\|_2 \cos(\boldsymbol{\theta}_{\psi(\boldsymbol{x}),k}) & \text{if } k = y, \\ p_k(\boldsymbol{x}) \|\psi(\boldsymbol{x})\|_2 \cos(\boldsymbol{\theta}_{\psi(\boldsymbol{x}),k}) & \text{if } k \neq y. \end{cases}
\end{aligned} \tag{11}$$

### A.1   NOTATION AND ASSUMPTIONS

We make the following assumptions, which are standard and realistic, serving as regular conditions for our theoretical analysis.

**Assumption 1** (Feature Extractor and Classifier Head). *Let $\psi : \mathcal{X} \to \mathbb{R}^p$ be the feature extractor, and let the classifier head be parameterized by $\{\mathbf{W}_k, b_k\}_{k=1}^C$. For an input $\boldsymbol{x}$, the class scores and softmax probabilities are*

$$s_k(\boldsymbol{x}) = \mathbf{W}_k^\top \psi(\boldsymbol{x}) + b_k, \qquad p_k(\boldsymbol{x}) = \frac{\exp(s_k(\boldsymbol{x}))}{\sum_{j=1}^C \exp(s_j(\boldsymbol{x}))}.$$

**Assumption 2** (Global Upper Bounds). *We assume the existence of global upper bounds:*

$$\sup_{\boldsymbol{x} \in \mathcal{X}} \|\psi(\boldsymbol{x})\|_2 \leq B_\psi, \qquad \sup_k \|\mathbf{W}_k\|_2 \leq B_w, \qquad \sup_{k \neq y} |b_y - b_k| \leq B.$$

**Assumption 3** (Small Cross-Entropy Condition). *Each training sample $(\boldsymbol{x}, y)$ satisfies a small cross-entropy condition:*

$$\ell_{ce}(f(\boldsymbol{x}), y) = -\log p_y(\boldsymbol{x}) \leq \varepsilon', \qquad \varepsilon' \ll 1. \tag{12}$$

**Assumption 4** (Margin Condition). *The small cross-entropy condition in Assumption 3 implies a positive margin:*

$$m_{\varepsilon'} := \log \frac{1 - \varepsilon'}{\varepsilon'} - \log(C - 1) > 0. \tag{13}$$

### A.2   PROPOSITIONS

**Proposition 2.** *Let $(\boldsymbol{x}, y)$ satisfy (12), and suppose the margin (13) dominates bias and norm terms, $m_{\varepsilon'} > B + B_w B_\psi$. Then the angle between the feature vector $\psi(\boldsymbol{x})$ and the correct weight vector $\mathbf{W}_y$ is upper-bounded by*

$$\theta_y(\boldsymbol{x}) \leq \arccos \frac{m_{\varepsilon'} - B - B_w B_\psi}{B_w B_\psi}. \tag{14}$$

*In particular, as $\varepsilon' \to 0$, $m_{\varepsilon'} \to \infty$ and $\theta_y(\boldsymbol{x}) \to 0$.*

*Proof.* From (12), $p_y(\boldsymbol{x}) \geq 1 - \varepsilon'$, which implies for all $k \neq y$,

$$s_y - s_k = (\mathbf{W}_y - \mathbf{W}_k)^\top \psi(\boldsymbol{x}) + (b_y - b_k) \geq m_{\varepsilon'}.$$

Subtracting the bias term gives

$$(\mathbf{W}_y - \mathbf{W}_k)^\top \psi(\boldsymbol{x}) \geq m_{\varepsilon'} - B.$$

Writing the inner product in cosine form, using $\|\mathbf{W}_y - \mathbf{W}_k\| \leq 2B_w$ and $\|\psi(\boldsymbol{x})\| \leq B_\psi$,

$$S_c(\mathbf{W}_y - \mathbf{W}_k, \psi(\boldsymbol{x})) \geq \frac{m_{\varepsilon'} - B}{2B_w B_\psi} > 0.$$

Finally, decompose $\mathbf{W}_y^\top \psi(\boldsymbol{x}) = (\mathbf{W}_y - \mathbf{W}_k + \mathbf{W}_k)^\top \psi(\boldsymbol{x})$ and apply the norm bounds:

$$\mathbf{W}_y^\top \psi(\boldsymbol{x}) \geq m_{\varepsilon'} - B - B_w B_\psi, \quad \cos\theta_y(\boldsymbol{x}) = \frac{\mathbf{W}_y^\top \psi(\boldsymbol{x})}{\|\mathbf{W}_y\|\|\psi(\boldsymbol{x})\|} \geq \frac{m_{\varepsilon'} - B - B_w B_\psi}{B_w B_\psi}.$$

This proves (14). $\qquad\square$

***Remark*** 3. Proposition 2 formalizes the intuition that, for a training sample with a sufficiently small cross-entropy loss, the corresponding feature vector $\psi(\boldsymbol{x})$ aligns closely with the weight vector of the correct class $\mathbf{W}_y$. Specifically, when the margin $m_{\varepsilon'}$ dominates the bias and norm terms, the angle $\theta_y(\boldsymbol{x})$ between $\psi(\boldsymbol{x})$ and $\mathbf{W}_y$ is tightly upper-bounded. As the cross-entropy loss approaches zero, the margin $m_{\varepsilon'}$ grows, causing $\theta_y(\boldsymbol{x})$ to approach zero. Intuitively, this means that highly confident predictions correspond to feature vectors that are nearly collinear with the correct class weight, which underpins the effectiveness of norm-based adjustments in class-wise robustness analysis.

### A.3 HARD-VS-EASY CLASSES: A FORMAL GAP INEQUALITY

**Notation** Let $(\boldsymbol{x}, y)$ denote a training sample, and let $\boldsymbol{x}^{\mathrm{adv}}$ be its adversarial counterpart generated within a perturbation budget $\varepsilon$. For a fixed class $k$, we define the *clean* and *adversarial* predictive probabilities as

$$p_k := p_k(\boldsymbol{x}), \quad p_k^{\mathrm{adv}} := p_k(\boldsymbol{x}^{\mathrm{adv}}).$$

For samples belonging to class $k$, the pointwise gap as defined in Definition 1 is given by

$$\delta(\boldsymbol{x}, y = k) := \left| \frac{\partial \ell_{\mathrm{ce}}(f(\boldsymbol{x}^{\mathrm{adv}}), y)}{\partial \|\mathbf{W}_y\|_2} \right| - \left| \frac{\partial \ell_{\mathrm{ce}}(f(\boldsymbol{x}), y)}{\partial \|\mathbf{W}_y\|_2} \right|$$
$$= \left( p_k - p_k^{\mathrm{adv}} \right) \|\psi(\boldsymbol{x})\|_2 \cos\theta_{\psi(\boldsymbol{x}),k} \text{ (by } Equation \text{ (11))}$$

In addition, we define the expected gradient gap as

$$\Delta_k := \mathbb{E}_{(\mathbf{X}, Y=k)} \delta(\mathbf{X}, Y)$$

.

We further introduce the shorthand notation

$$Z(\boldsymbol{x}, k) := \|\psi(\boldsymbol{x})\|_2 \cos\theta_{\psi(\boldsymbol{x}),k},$$

and note that Assumption A.1 ensures

$$Z(\boldsymbol{x}^{\mathrm{adv}}, k) \simeq Z(\boldsymbol{x}, k) \tag{15}$$

for adversarially robust trained model [24; 26].

**Assumption 5** (Feature–angle stationarity)**.** *For every class $k$, the random variable $Z(\mathbf{X}, k)$ is independent of $\left( p_k, p_k^{adv} \right)$ and has finite mean $\mu_Z := \mathbb{E}[Z(\mathbf{X}, k)] > 0$.*

***Remark*** 4. The assumption $\mu_Z > 0$ is practically necessary to ensure a meaningful interpretation of the gap measure: if $\mu_Z$ were zero or negative, the relationship between class hardness and gradient gaps would become inverted or trivial, violating the intuitive notion of robustness and class difficulty alignment. However, well-trained neural networks typically satisfy $\mu_Z > 0$, validating our assumption.

**Proposition 1.** $\Delta_k = \mu_Z H_k$ *holds. Consequently, if a class $c_{hard}$ is harder than class $c_{easy}$ ($H_{c_{hard}} > H_{c_{easy}}$), then $\Delta_{c_{hard}} > \Delta_{c_{easy}}$.*

*Proof.* Condition on $y = k$, $\Delta_k = \mathbb{E}_{(\mathbf{X}, Y=k)}\big[(p_k - p_k^{\text{adv}})\, Z(\mathbf{X}, Y)\big]$. By Assumption 5, $Z(\boldsymbol{X}, k)$ is independent of $(p_k - p_k^{\text{adv}})$ and shares the same distribution for all samples of class $k$. Hence the expectation factorizes:

$$\Delta_k = \mathbb{E}_{(\mathbf{X}, Y=k)}[p_k - p_k^{\text{adv}}] \,\cdot\, \mathbb{E}_{(\mathbf{X}, Y=k)}[Z(\mathbf{X}, Y)] = H_k\, \mu_Z.$$

Because $\mu_Z > 0$, the ordering of $\Delta_k$ follows directly from the ordering of $H_k$. $\qquad\square$

**Notation**  Let the *clean* and *adversarial* per-sample gradients for class $k$ be

$$g_k^{\text{cln}}(\boldsymbol{x}, y) := \frac{\partial \ell_{\text{ce}}\big(f(\boldsymbol{x}), y\big)}{\partial \mathbf{W}_k}, \quad g_k^{\text{adv}}(\boldsymbol{x}, y) := \frac{\partial \ell_{\text{ce}}\big(f(\boldsymbol{x}^{\text{adv}}), y\big)}{\partial \mathbf{W}_k}, \tag{16}$$

and denote the unit direction $\widetilde{\mathbf{W}}_k := \mathbf{W}_k / \|\mathbf{W}_k\|_2$. Define the scalar projections

$$s^{\text{cln}}(\boldsymbol{x}, y) := \widetilde{\mathbf{W}}_k^\top g_k^{\text{cln}}(\boldsymbol{x}, y), \quad s^{\text{adv}}(\boldsymbol{x}, y) := \widetilde{\mathbf{W}}_k^\top g_k^{\text{adv}}(\boldsymbol{x}, y). \tag{17}$$

The sample-wise gap from Definition 1 is $\delta(\boldsymbol{x}, y) = |s^{\text{adv}}| - |s^{\text{cln}}|$.

**Lemma 1.** *For every SGD iteration $t$,*

$$\mathbb{E}_{(\mathbf{X}, Y)\sim\mathcal{D}}\big[\widetilde{\mathbf{W}}_k^{(t)\top} g_k^{(t)}\big] = -\Delta_k, \quad \text{where } \Delta_k := \mathbb{E}_{(\mathbf{X}, Y=k)}[\delta(\mathbf{X}, k)].$$

*Proof.* We prove the lemma by separating the contributions from samples of class $k$ and non-target classes.

**Case 1: $Y \neq k$.** For samples not belonging to class $k$, the indicator $\mathbf{1}\{k = Y\} = 0$, so both clean and adversarial class-$k$ scores are positive:

$$s^{\text{cln}} > 0, \quad s^{\text{adv}} > 0.$$

PGD perturbations primarily target the true class $Y$, leaving non-target class logits largely unchanged. Hence,

$$s^{\text{adv}} \approx s^{\text{cln}} \quad \Rightarrow \quad \delta(\mathbf{X}, Y) = 0.$$

These samples therefore contribute positively to the inner product $\widetilde{\mathbf{W}}_k^\top g^{\text{adv}}$, but they do not contribute to the expected gradient gap $\Delta_k$. In other words, for samples whose true label is not $k$, the adversarial perturbation has little effect on the gradient gap because the model is already unlikely to predict class $k$.

**Case 2: $Y = k$.** For samples of class $k$, the clean logit is high, $p_k(\mathbf{X}) \approx 1$, so $s^{\text{cln}} \approx 0$. Adversarial perturbations decrease this logit significantly, $p_k(\mathbf{X}^{\text{adv}}) \ll 1$, giving $s^{\text{adv}} < 0$. Consequently, the gradient gap satisfies

$$\delta(\mathbf{X}, k) = |s^{\text{adv}}| - |s^{\text{cln}}| = -s^{\text{adv}}(\mathbf{X}, k).$$

Thus, for these samples,

$$\widetilde{\mathbf{W}}_k^\top g^{\text{adv}} = -\delta(\mathbf{X}, k).$$

**Combine the two cases.** Taking the expectation over the data distribution $\mathcal{D}$, we have

$$\begin{aligned}
\mathbb{E}_{(\mathbf{X}, Y)}[\widetilde{\mathbf{W}}_k^\top g_k^{\text{adv}}] &= \mathbb{E}_{Y\neq k}[\widetilde{\mathbf{W}}_k^\top g_k^{\text{adv}}] + \mathbb{E}_{Y=k}[\widetilde{\mathbf{W}}_k^\top g_k^{\text{adv}}] \\
&= 0 + \mathbb{E}_{Y=k}[-\delta(\mathbf{X}, k)] \\
&= -\Delta_k.
\end{aligned}$$

Hence, the expected projected adversarial gradient is exactly $-\Delta_k$, which governs the average change of the head norm in Theorem 1. $\qquad\square$

**Theorem 1.** *Run stochastic gradient descent with learning rate $\eta$ for $T$ iterations using the adversarial loss. Let $\Delta_k := \mathbb{E}_{(\mathbf{X}, Y=k)}\delta(\mathbf{X}, Y)$ be the class-specific expected gradient gap. Then,*

$$\mathbb{E}\|\mathbf{W}_k^{(T)}\|_2 = \|\mathbf{W}_k^{(0)}\|_2 + \eta T \Delta_k. \tag{5}$$

*Consequently, if a class $c_{hard}$ is* harder *than a class $c_{easy}$ ( $\Delta_{c_{hard}} > \Delta_{c_{easy}}$ ), there exists $T^*$ such that $\mathbb{E}\|\mathbf{W}_{c_{hard}}^{(T)}\|_2 > \mathbb{E}\|\mathbf{W}_{c_{easy}}^{(T)}\|_2$ for all $T \geq T^*$.*

*Proof.* Let $g_k^{\mathrm{adv},(t)} := \dfrac{\partial \ell_{\mathrm{ce}}\big(f(\boldsymbol{x}^{\mathrm{adv}}), y\big)}{\partial \mathbf{W}_k^{(t)}}$ denote the stochastic gradient at iteration $t$ with adversarial loss. One SGD step updates $\mathbf{W}_k^{(t+1)} = \mathbf{W}_k^{(t)} - \eta\, g_k^{\mathrm{adv},(t)}$.

We are interested only in the *change of the norm* $\|\mathbf{W}_k\|_2$ and not in the change of its direction. Decompose the gradient into a part parallel to $\mathbf{W}_k^{(t)}$ and an orthogonal part:

$$g_k^{\mathrm{adv},(t)} = \big(\widetilde{\mathbf{W}}_k^{(t)\top} g_k^{\mathrm{adv},(t)}\big)\, \widetilde{\mathbf{W}}_k^{(t)} + \big[g_k^{\mathrm{adv},(t)} - (\widetilde{\mathbf{W}}_k^{(t)\top} g_k^{\mathrm{adv},(t)})\widetilde{\mathbf{W}}_k^{(t)}\big],$$

where $\widetilde{\mathbf{W}}_k^{(t)} := \mathbf{W}_k^{(t)}/\|\mathbf{W}_k^{(t)}\|_2$ is the unit vector in the current direction. Only the **parallel component** $\big(\widetilde{\mathbf{W}}_k^{(t)\top} g_k^{\mathrm{adv},(t)}\big)\widetilde{\mathbf{W}}_k^{(t)}$ can increase or decrease the *length*; the orthogonal component merely rotates $\mathbf{W}_k^{(t)}$ and leaves its norm unchanged to first order.

Formally,

$$\|\mathbf{W}_k^{(t+1)}\|_2^2 = \big\|\mathbf{W}_k^{(t)} - \eta g_k^{\mathrm{adv},(t)}\big\|_2^2 = \|\mathbf{W}_k^{(t)}\|_2^2 - 2\eta\, \widetilde{\mathbf{W}}_k^{(t)\top} g_k^{\mathrm{adv},(t)}\, \|\mathbf{W}_k^{(t)}\|_2 + \eta^2 \|g_k^{\mathrm{adv},(t)}\|_2^2.$$

Ignoring the $O(\eta^2)$ term (standard in first-order SGD analysis) and taking square roots yields

$$\|\mathbf{W}_k^{(t+1)}\|_2 \approx \|\mathbf{W}_k^{(t)}\|_2 - \eta\, \widetilde{\mathbf{W}}_k^{(t)\top} g_k^{\mathrm{adv},(t)}.$$

Hence we *project the gradient onto* $\widetilde{\mathbf{W}}_k^{(t)}$ because that scalar product $\widetilde{\mathbf{W}}_k^{(t)\top} g_k^{\mathrm{adv},(t)}$ is the *exact* first-order change in the norm of $\mathbf{W}_k$.

**Taking expectations.** **Lemma 1** induces $\mathbb{E}_{(\mathbf{X},Y)}[\widetilde{\mathbf{W}}_k^{(t)\top} g_k^{\mathrm{adv},(t)}] = -\Delta_k$ for every iteration.[1] Therefore,

$$\mathbb{E}\|\mathbf{W}_k^{(t+1)}\|_2 = \mathbb{E}\|\mathbf{W}_k^{(t)}\|_2 + \eta\, \Delta_k.$$

Unrolling the recursion over $T$ steps gives (5).

**Hard vs. Easy classes.** If $\Delta_{c_{\mathrm{hard}}} > \Delta_{c_{\mathrm{easy}}}$ (by **Proposition 1**), their expected norm difference grows as $\eta\, T(\Delta_{c_{\mathrm{hard}}} - \Delta_{c_{\mathrm{easy}}})$, so after $T^* := \big(\|\mathbf{W}_{c_{\mathrm{easy}}}^{(0)}\|_2 - \|\mathbf{W}_{c_{\mathrm{hard}}}^{(0)}\|_2\big)\big/\big[\eta(\Delta_{c_{\mathrm{hard}}} - \Delta_{c_{\mathrm{easy}}})\big]$ the inequality $\mathbb{E}\|\mathbf{W}_{c_{\mathrm{hard}}}^{(T)}\|_2 > \mathbb{E}\|\mathbf{W}_{c_{\mathrm{easy}}}^{(T)}\|_2$ holds for all $T \geq T^*$. $\qquad\square$

**Proposition 3.** *Under Assumptions A.1, we have $\delta(\boldsymbol{x}, y) > \rho - \epsilon''$ for $\rho > 0$ and small $\epsilon'' > 0$ for every pair $(\boldsymbol{x}, y, \boldsymbol{x}^{adv})$.*

*Proof.* For class $k$, the chain rule gives

$$\frac{\partial \ell_{\mathrm{ce}}(f(\boldsymbol{x}), y)}{\partial s_k(\boldsymbol{x})} = p_k(\boldsymbol{x}) - \mathbf{1}\{k = y\}, \quad \frac{\partial s_k(\boldsymbol{x})}{\partial \|\mathbf{W}_k\|_2} = \|\psi(\boldsymbol{x})\|_2 \cos\theta_{\psi(\boldsymbol{x}),k}.$$

For the true class $k = y$, combining these yields

$$g(\boldsymbol{x}, y) := \frac{\partial \ell_{\mathrm{ce}}(f(\boldsymbol{x}), y)}{\partial \|\mathbf{W}_y\|_2} = (1 - p_y(\boldsymbol{x}))\, \|\psi(\boldsymbol{x})\|_2 \cos\theta_{\psi(\boldsymbol{x}),y} := (1 - p_y(\boldsymbol{x}))\, Z_{\boldsymbol{x},y}. \tag{18}$$

---

[1]The unit vector $\widetilde{\mathbf{W}}_k^{(t)}$ is independent of the minibatch sampled at step $t$, so we may pull it outside the expectation.

Define the adversarial gradient gap

$$\delta(\boldsymbol{x}, y) = (1 - p_y(\boldsymbol{x}^{\mathrm{adv}}))Z_{\mathrm{adv}} - (1 - p_y(\boldsymbol{x}))Z_{\mathrm{cln}},$$

where $Z_{\mathrm{adv}} = Z_{\boldsymbol{x}^{\mathrm{adv}}, y}$ and $Z_{\mathrm{cln}} = Z_{\boldsymbol{x}, y}$. This can be rewritten as

$$\delta(\boldsymbol{x}, y) = \underbrace{(p_y(\boldsymbol{x}) - p_y(\boldsymbol{x}^{\mathrm{adv}}))Z_{\mathrm{cln}}}_{:=(A)} + \underbrace{(1 - p_y(\boldsymbol{x}^{\mathrm{adv}}))(Z_{\mathrm{adv}} - Z_{\mathrm{cln}})}_{:=(B)}.$$

Part (A)

Since the adversarial attack reduces the probability of the true class, $p_y(\boldsymbol{x}^{\mathrm{adv}}) < p_y(\boldsymbol{x})$, and by training $Z_{\mathrm{cln}} > 0$, the first term $(p_y(\boldsymbol{x}) - p_y(\boldsymbol{x}^{\mathrm{adv}}))Z_{\mathrm{cln}}$ is strictly positive. Denote the positive magnitude by $\rho > 0$.

Part (B)

Under Assumptions A.1, Eq. (15) can be approximated. Then, the change in the cosine term is small: $|Z_{\mathrm{adv}} - Z_{\mathrm{cln}}| \leq \epsilon''$, and $1 - p_y(\boldsymbol{x}^{\mathrm{adv}}) \leq 1$. Hence the second term satisfies

$$(1 - p_y(\boldsymbol{x}^{\mathrm{adv}}))(Z_{\mathrm{adv}} - Z_{\mathrm{cln}}) \geq -\epsilon''.$$

Part (A) + (B)

Adding the positive and negative parts gives

$$\delta(\boldsymbol{x}, y) \geq (p_y(\boldsymbol{x}) - p_y(\boldsymbol{x}^{\mathrm{adv}}))Z_{\mathrm{cln}} - \epsilon'' = \rho - \epsilon''.$$

Thus, the adversarial gradient gap is lower-bounded by $\rho - \epsilon''$, as claimed. $\qquad\square$

***Remark*** 5. Proposition 3 guarantees that for each training sample $(\boldsymbol{x}, y)$, the adversarial gradient gap $\delta(\boldsymbol{x}, y)$ is lower-bounded by $\rho - \epsilon''$. Consequently, the class-specific expected gradient gap $\Delta_k = \mathbb{E}_{(\mathbf{X}, Y=k)}[\delta(\mathbf{X}, Y)]$ remains stable and mostly positive. Moreover, Theorem 1 shows that the norm of the classifier head $\mathbf{W}_k$ evolves approximately linearly with $\Delta_k$:

$$\mathbb{E}\|\mathbf{W}_k^{(T)}\|_2 = \|\mathbf{W}_k^{(0)}\|_2 + \eta T \Delta_k.$$

Since harder classes have larger expected gradient gaps ($\Delta_{C_{\mathrm{hard}}} > \Delta_{C_{\mathrm{easy}}}$), their corresponding head norms grow more rapidly during training. These observations collectively provide an intuitive explanation: the stability of $\delta(\boldsymbol{x}, y)$ ensures that the head norm growth predicted by Theorem 1 is reliable. Consequently, harder classes naturally acquire larger norms, resulting in stronger logits and improved robustness, while easier classes remain relatively balanced.

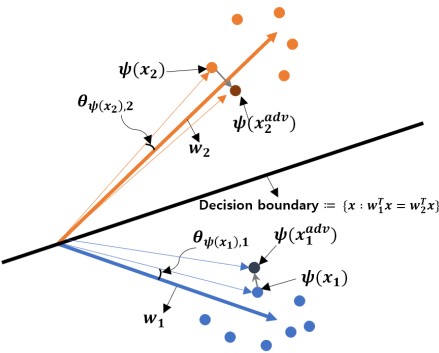

Figure 4: **Small Loss Scenario**. Consider the case w/o bias term for simplicity. Orange and blue circles are data point in feature space of class 1 and 2, respectively. If train loss is small and robust features well trained, $\theta_{\psi(\boldsymbol{x}),y}$ and $\theta_{\psi(\boldsymbol{x}^{\mathrm{adv}}),y}$ are small.

Figure 4 illustrates a scenario where the loss becomes small through adversarial training.

## B  ALGORITHM

---
**Algorithm 1:** DecoupledSAM (DecoSAM)

---
**Input** : $\psi$ : feature extractor, $\mathbf{W}$ : weight of head, $\boldsymbol{b}$ : bias of head, $s$ : standard trained model, dataset
$\qquad \mathcal{D} = \{(\boldsymbol{x}_i, y_i)\}_{i=1}^{n}$ , number of epochs $T$, perturbation budget $\varepsilon$, number of batch $B$, batch size
$\qquad J$, adversarial training algorithm $\mathcal{A}$

**Output:** adversarially robust network

1  $\psi, \mathbf{W}, \boldsymbol{b}$ (train the adversarially robust model)
2  Freeze $\psi$ and $\boldsymbol{b}$
3  **for** $b = 1, \cdots, B$ **do**
4  $\quad$ Compute $\nu_k$ and $\rho_k = \rho * \nu_k$ for all $k$.
5  $\quad$ $\mathbf{W} \leftarrow \widetilde{\mathbf{W}}(HWNwB)$
6  $\quad$ **for** $j = 1, \cdots, J$ **do**
7  $\quad\quad$ Generate $\widehat{\boldsymbol{x}}_j^{\mathrm{adv}}$ by PGD($W \circ \psi(\boldsymbol{x}_j), y_j$),
8  $\quad\quad$ Update $\mathbf{W}$ with DecoSAM($\mathcal{A}(\widehat{\boldsymbol{x}}_j^{\mathrm{adv}}, y_j)$) in (7)
9  $\quad$ **end**
10  **end**
11  **Return** $\mathbf{W}, \psi$

---

## C  A VALIDATION SET-FREE APPROACH

In situations where labeled data are limited or labeling is costly-such as requiring expert input or facing privacy concerns [29] - using a separate validation set reduces the data available for training or necessitates additional labeling effort, both of which can harm model performance [23; 21; 9]. By eliminating the need for a validation set, our approach fully utilizes the limited labeled data for training, making it more practical and cost-effective in real-world scenarios.

## D  EXPERIMENTAL DETAILS

**Common**   We follow the experimental setting of [15] for our study. In pre-training phase, we train various adversarial robust learning algorithms (PGD-AT, TRADES, MART, and ARoW) using SGD optimizer with a momentum of 0.9 and weight decay of $5e^{-4}$. To mitigate robust overfitting, we implement a multi-step learning rate scheduler that reduces the learning rate by a factor of 0.1 at epochs 90 and 95, and select the model from the final epoch without using a validation set. In DecoSAM stage, we employ an SGD optimizer with a momentum of 0.9, a learning rate of 0.01,

a batch size of 512, and a perturbation size $\rho = 5e - 5$. Our experiments are conducted using an NVIDIA RTX 3090 GPU with 24 GB of memory.

**STL-10** STL-10 dataset is a benchmark designed for evaluating supervised and semi-supervised learning algorithms, particularly in scenarios with limited labeled data. It consists of 96 ×96 pixel images across 10 classes, with 5,000 labeled training images, 8,000 labeled test images, and 100,000 unlabeled images from a broader distribution. Its focus on small labeled datasets and abundant unlabeled data makes it ideal for testing algorithms that aim to learn robust features or leverage unlabeled data effectively. It has higher resolution than CIFAR-10.

For our experiments on STL-10, we employ a two-stage approach. We first train a teacher model using supervised learning on the labeled data, then utilized this teacher model to generate pseudo labels for the unlabeled data. Finally, we apply various adversarial training algorithms using both the labeled data and the pseudo-labeled unlabeled data.

# E  ADDITIONAL EXPERIMENTS

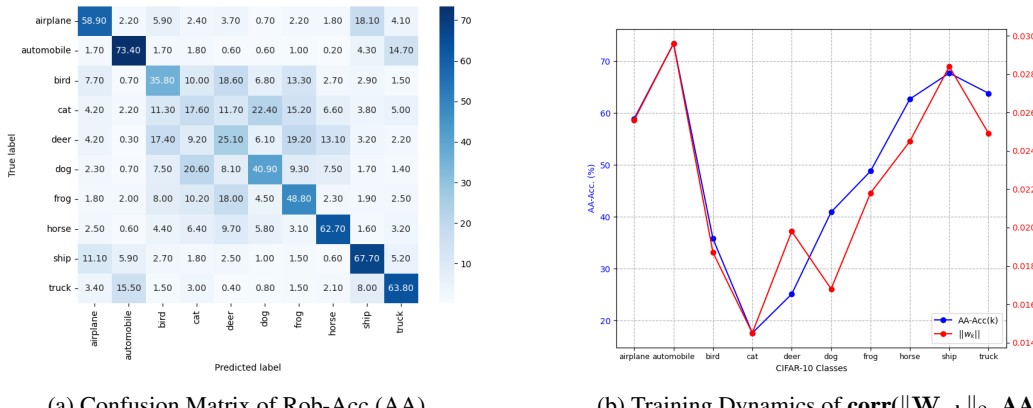

(a) Confusion Matrix of Rob-Acc (AA).

(b) Training Dynamics of $\mathbf{corr}(\|\mathbf{W_{rob}}\|_2, \mathbf{AA})$

Figure 5: Confusion matrix for AA and the correlation between head weights and robust accuracy against AA for a model trained using PGD-AT.

We provide the confusion matrix and the correlation between head weights and robust accuracy against PGD-20 in the manuscript. Figures5a and 5b demonstrate that similar patterns observed in PGD-20 are also present in AA. A notable feature is the heightened vulnerability of the class with the lowest robust accuracy, which becomes even more pronounced under AA.

**CIFAR100** CIFAR-100 is an extension of the CIFAR-10 dataset, designed to provide a more challenging classification task. While CIFAR-10 consists of 10 classes with 6,000 images per class, CIFAR-100 includes 100 classes with only 600 images per class, making the dataset more complex and less balanced. Each image in CIFAR-100, like CIFAR-10, is a 32x32 color image, but the increased number of classes and fewer samples per class require models to have greater capacity to generalize effectively. This makes CIFAR-100 particularly useful for evaluating algorithms in scenarios with fine-grained classification and limited training data per class. CIFAR-100, with its 100 classes and only 600 images per class, poses a more complex classification challenge compared to CIFAR-10. However, due to its large number of classes, it is not typically used as a benchmark dataset for robust fairness studies [18; 20; 28; 11]. This is because robust fairness often focuses on addressing disparities across a smaller set of classes, where the class-wise performance can be more effectively analyzed and compared. The high number of classes in CIFAR-100 makes it less suitable for such targeted evaluations. Table 6 reveals that the robust accuracy of the worst class against AA is significantly low. Therefore, in this scenario, it is advisable to consider the worst-class robust accuracy alongside other metrics for a more comprehensive evaluation. Across all algorithms and varying levels of model complexity, HWNwB demonstrates substantial improvements in robust fairness. Furthermore, DecoSAM maintains the robust fairness performance achieved by HWNwB while also enhancing overall robustness, showcasing its effectiveness.

Table 6: **Comparison of HWNwB and DecoSAM Performance on Baseline Algorithms on CIFAR-100. PGD** and **AA** indicates the robust accuracy under a 20-step PGD attack and the AutoAttack, respectively. **WC** indicates the worst-class robust accuracy, **STD** indicates the standard deviation of class-wise robust accuracies, and **Max-Min** indicates the difference between the highest and lowest class-wise robust accuracies.

| Method | Clean(↑) | PGD(↑) | WC(↑) | STD(↓) | Max-Min(↓) | corr($\|W\|_2$, PGD) | AA(↑) | WC(↑) | STD(↓) | Max-Min(↓) |
|---|---|---|---|---|---|---|---|---|---|---|
| | | | | | CIFAR-100 (WRN-28-2) | | | | | |
| PGD-AT | **53.16** | 26.90 | 0.00 | 19.10 | 74.00 | 0.8427 | **22.52** | 0.00 | 19.10 | 71.00 |
| + HWNwB | 50.30 | 26.76 | **2.00** | **14.64** | **64.00** | | 19.43 | 0.00 | **14.54** | 64.00 |
| + DecoSAM | 51.12 | **27.75** | 1.00 | 15.47 | 69.00 | | 20.34 | 0.00 | 15.01 | 61.00 |
| TRADES | **53.28** | 27.11 | 0.00 | 18.57 | 71.00 | 0.6632 | **22.13** | 0.00 | 18.20 | 71.00 |
| + HWNwB | 52.27 | 28.20 | **1.00** | 16.95 | 72.00 | | 20.80 | 0.00 | 16.42 | 70.00 |
| + DecoSAM | 52.05 | **28.29** | **1.00** | **16.70** | **69.00** | | 21.41 | 0.00 | **16.26** | **67.00** |
| MART | **48.93** | **28.63** | 0.00 | 19.39 | 72.00 | 0.8824 | **22.77** | 0.00 | 19.97 | 69.00 |
| + HWNwB | 44.20 | 25.42 | 2.00 | **14.67** | **61.00** | | 18.97 | 0.00 | **14.03** | **59.00** |
| + DecoSAM | 45.60 | 27.55 | **3.00** | 15.11 | 63.00 | | 20.16 | 0.00 | 15.84 | 62.00 |
| ARoW | **52.55** | 27.10 | 0.00 | 18.24 | 72.00 | 0.7002 | **22.42** | 0.00 | 18.30 | 71.00 |
| + HWNwB | 51.65 | 28.40 | **2.00** | **16.86** | **71.00** | | 21.33 | 0.00 | **16.33** | 68.00 |
| + DecoSAM | 51.66 | **28.56** | **2.00** | 16.95 | **71.00** | | 21.78 | 0.00 | 16.41 | 68.00 |
| | | | | | CIFAR-100 (WRN-28-5) | | | | | |
| PGD-AT | **61.03** | 30.60 | 0.00 | 18.52 | 74.00 | 0.7742 | **25.90** | 0.00 | 18.56 | 71.00 |
| + HWNwB | 59.50 | 32.43 | **6.00** | **15.94** | **68.00** | | 24.49 | 1.00 | **15.96** | **69.00** |
| + DecoSAM | 59.77 | **32.80** | **6.00** | 16.02 | 69.00 | | 25.13 | 1.00 | 16.30 | **69.00** |
| TRADES | **58.44** | 30.85 | 2.00 | 18.85 | 70.00 | 0.5697 | 25.99 | 1.00 | 18.96 | 73.00 |
| + HWNwB | 57.89 | 32.00 | 4.00 | 17.60 | 71.00 | | 25.57 | 2.00 | 17.78 | **69.00** |
| + DecoSAM | 57.32 | **32.96** | 4.00 | **14.74** | **70.00** | | **26.03** | 2.00 | 17.97 | 72.00 |
| MART | **56.42** | 32.68 | 0.00 | 19.16 | 76.00 | 0.8363 | **26.87** | 0.00 | 19.58 | 74.00 |
| + HWNwB | 53.19 | 32.02 | **6.00** | **15.71** | **62.00** | | 24.59 | 1.00 | **15.82** | **64.00** |
| + DecoSAM | 54.15 | **33.13** | **6.00** | 16.17 | 66.00 | | 25.72 | 1.00 | 16.68 | 69.00 |
| ARoW | **58.39** | 31.10 | 1.00 | 18.83 | 73.00 | 0.5918 | **26.60** | 1.00 | 18.84 | 70.00 |
| + HWNwB | 57.61 | 33.47 | **3.00** | 17.76 | 71.00 | | 25.86 | 1.00 | 17.53 | **66.00** |
| + DecoSAM | 57.77 | **33.54** | **3.00** | **17.44** | **69.00** | | 26.20 | 1.00 | **16.95** | 67.00 |
| | | | | | CIFAR-100 (WRN-28-10) | | | | | |
| PGD-AT | **63.94** | 29.11 | 2.00 | 17.49 | 69.00 | 0.7137 | **26.29** | 2.00 | 17.62 | 68.00 |
| + HWNwB | 63.19 | 33.47 | 8.00 | **16.07** | **64.00** | | 25.48 | 4.00 | **16.25** | 65.00 |
| + DecoSAM | 62.29 | **33.68** | **9.00** | 16.38 | **64.00** | | 26.01 | **5.00** | 16.71 | **64.00** |
| TRADES | **60.01** | 31.85 | 5.00 | 18.21 | 70.00 | 0.4811 | **27.53** | 3.00 | 18.44 | 72.00 |
| + HWNwB | 59.40 | 34.42 | **5.00** | **17.67** | **69.00** | | 27.04 | **4.00** | 17.62 | **68.00** |
| + DecoSAM | 59.44 | **34.64** | **5.00** | 17.84 | 70.00 | | 27.46 | **4.00** | **17.58** | **68.00** |
| MART | **59.86** | 32.51 | 3.00 | 18.40 | 70.00 | 0.8137 | 27.78 | 0.00 | 18.78 | 72.00 |
| + HWNwB | 57.74 | 34.12 | **7.00** | **15.48** | **63.00** | | 26.69 | 1.00 | **16.07** | 68.00 |
| + DecoSAM | 57.87 | **34.55** | **7.00** | 16.35 | 66.00 | | 27.35 | 1.00 | 16.78 | **67.00** |
| ARoW | **59.30** | 31.44 | 3.00 | 17.90 | **69.00** | 0.4892 | 27.58 | 3.00 | 18.38 | 68.00 |
| + HWNwB | 59.09 | 34.71 | **4.00** | **17.37** | **69.00** | | 27.06 | 3.00 | **17.55** | **66.00** |
| + DecoSAM | 58.58 | **34.81** | **4.00** | 17.97 | 72.00 | | **27.75** | **4.00** | 17.98 | 68.00 |

**OfficeHome** Table 7 reports the adversarial performance of various baseline methods and their combinations with HWNwB and DecoSAM on the OfficeHome real-world domain using ResNet-50. We evaluate models in terms of clean accuracy (Clean), average accuracy across all classes (AA), worst-class accuracy (WC), and the average accuracy of the lowest 5% of classes (WC(5%)). Across the PGD-AT, TRADES, and ARoW baselines, incorporating DecoSAM consistently improves AA, WC, and WC(5%), indicating that DecoSAM effectively enhances robustness for both typical and hard-to-classify classes. Notably, DecoSAM yields the highest WC(5%) in all three baseline blocks, suggesting that it particularly benefits the most vulnerable classes. The combination with HWNwB also improves WC in many cases, though DecoSAM generally achieves stronger overall gains. FAAL is included as a reference robust fairness method, and while its WC is competitive, DecoSAM applied to other baselines demonstrates superior balance between average and worst-class performance.

Table 7: **Adversarial performance on the OfficeHome real-world domain (ResNet-50, 65 classes).** WC(5%) denotes the average accuracy of the lowest 5% classes (3 classes). Best values in each block are bolded.

| Method | Clean (↑) | AA (↑) | WC (↑) | WC(5%) (↑) |
|---|---|---|---|---|
| PGD-AT | 93.59 | 87.76 | 65.57 | 72.15 |
| +HWNwB | 93.32 | 87.54 | 68.57 | 74.36 |
| +DecoSAM | **93.68** | **87.95** | **68.84** | **74.80** |
| TRADES | 93.32 | 86.50 | 60.61 | 66.64 |
| +HWNwB | 93.36 | 86.82 | 63.64 | 70.65 |
| +DecoSAM | **93.66** | **86.65** | **63.97** | **70.88** |
| ARoW | **93.89** | 87.15 | 66.57 | 68.14 |
| +HWNwB | 93.84 | 86.36 | 67.90 | 70.46 |
| +DecoSAM | 93.80 | **86.54** | **68.11** | **70.96** |
| FAAL | 92.41 | 86.01 | 62.57 | 68.44 |

### E.1 Comparison DecoSAM + ARoW to FAAL

Table 8: **Comparison of ARoW+DecoSAM with FAAL across datasets.** We report Clean accuracy, Average Accuracy (AA), and Worst-Class accuracy (WC).

| Dataset | Method | Clean (↑) | AA (↑) | WC (↑) |
|---|---|---|---|---|
| CIFAR10 | FAAL | 81.19 | 48.81 | 32.80 |
| CIFAR10 | ARoW + DecoSAM | **83.18** | **48.98** | **34.70** |
| CIFAR100 | FAAL | 55.51 | 25.66 | **1.00** |
| CIFAR100 | ARoW + DecoSAM | **57.77** | **26.20** | **1.00** |
| STL10 | FAAL | 78.87 | 58.44 | 32.66 |
| STL10 | ARoW + DecoSAM | **80.22** | **59.58** | **35.71** |

Table 8 compares the performance of ARoW+DecoSAM with FAAL on three datasets: CIFAR10, CIFAR100, and STL10. Across all datasets, ARoW+DecoSAM consistently improves the clean accuracy and average accuracy (AA). Additionally, it enhances the worst-class accuracy (WC) in CIFAR10 and STL10, indicating that DecoSAM effectively mitigates class-wise disparity while maintaining overall robustness. For CIFAR100, WC remains very low, reflecting the inherent difficulty of some classes, yet ARoW+DecoSAM still slightly improves AA, showing its benefit even under challenging scenarios.

### E.2 Ablation Studies

In this section, we provide additional ablation studies examining the effects of robust regularization intensity as well as the influence of $\rho$ in DecoSAM.

### E.3 EFFECT OF ROBUST REGULARIZATION INTENSITY

We conduct experiments by varying the robust regularization parameter in TRADES. The surrogate version of the robust risk in TRADES is as follows:

$$\frac{1}{n} \sum_{i=1}^{n} \{\ell_{\text{ce}}(f_{\boldsymbol{\theta}}(\boldsymbol{x}_i), y_i) + \lambda \, \mathrm{D}_{\text{KL}}(\boldsymbol{p}_{\boldsymbol{\theta}}(\boldsymbol{x}_i) \| \boldsymbol{p}_{\boldsymbol{\theta}}(\boldsymbol{x}_i^{\text{adv}}))\} \tag{19}$$

where $\mathrm{D}_{\text{KL}}$ denotes the KL-divergence and $\lambda$ is the robust regularization parameter that controls the trade-off between generalization and robustness.

Table 9 shows that as increasing $\lambda$, the norms of $\mathbf{W}$ tend to exhibit a stronger correlation between class-wise robust accuracies. Additionally, for the worst-class robust accuracies, we observe an improvement in overall robust accuracy, suggesting that improving the overall robust accuracy also benefits worst-class performance. This implies that methods like FAAL [28], which focus on fine-tuning after training with a robust approach, or our proposed method, offer new directions for enhancing worst-class robustness.

Table 9: **Effect of Robust Regularization in TRADES.**

| $\lambda$ | CIFAR-10 (WRN-28-5) | | | |
|---|---|---|---|---|
| | corr($\|\mathbf{W}\|_2$, PGD) | Clean | PGD | WC |
| 0.5 | 0.5805 | 88.74 | 42.62 | 12.50 |
| 1 | 0.8176 | 88.43 | 48.32 | 18.70 |
| 2 | 0.8825 | 87.20 | 51.16 | 24.60 |
| 4 | 0.9191 | 85.07 | 53.34 | 27.80 |
| 6 | 0.9135 | 82.94 | 53.51 | 29.20 |
| 8 | 0.9230 | 81.83 | 53.76 | 29.20 |
| 10 | 0.9188 | 80.67 | 53.53 | 28.90 |

### E.4 EFFECT OF $\rho$ IN DECOSAM

Table 10 presents the effect of the hyperparameter $\rho$ in DecoSAM on class-wise adversarial performance on CIFAR-10. Using PGD-AT [12] as the base adversarial training algorithm, we vary $\rho$ and observe that both average accuracy (AA) and worst-class accuracy (WC) exhibit non-trivial changes. Values of $\rho$ within the narrow range of 0.00003–0.00011 maintain a reasonable level of AA, while WC fluctuates more substantially, indicating that $\rho$ primarily influences the model's robustness for harder classes. Specifically, beyond a certain threshold(0.00006) of $\rho$, we observe a trade-off: increasing $\rho$ tends to further improve overall accuracy (AA) while degrading worst-class accuracy (WC). The baseline configuration corresponding to $\rho = 0$ (HWNwB) yields lower AA but maintains a moderate WC, suggesting that DecoSAM with a properly tuned $\rho$ can improve both overall and worst-class performance simultaneously.

Table 10: **Effect of $\rho$ in DecoSAM.**

| $\rho$ | AA ($\uparrow$) | WC ($\uparrow$) |
|---|---|---|
| 0.00011 | **48.03** | 20.22 |
| 0.00009 | 47.92 | 20.14 |
| 0.00007 | 47.84 | 22.80 |
| 0.00006 | 47.93 | **25.10** |
| 0.00005 | 47.57 | 24.11 |
| 0.00003 | 47.96 | 21.46 |
| 0 (HWNwB) | 46.45 | 22.27 |

## F NEW EXPERIMENTAL RESULTS

### F.1 EFFECT OF $\tau$ IN DECOSAM

We set the default value to $\tau = 1.0$ primarily for simplicity, and the $\tau$-sweep results confirm that this choice aligns well with the underlying intuition. As shown in Eq. (8), increasing $\tau$ sharpens

Table 11: Effect of the temperature parameter $\tau$ on CIFAR-10 robustness (PGD-AT + DecoSAM, WRN-28-5, $\varepsilon = 8/255$).

| $\tau$ | Clean | PGD | WC(PGD) | AA | WC(AA) |
|---|---|---|---|---|---|
| 0.1 | 85.73 | 56.05 | 29.40 | 49.35 | 22.60 |
| 0.2 | 85.61 | 56.22 | 32.10 | 49.18 | 24.30 |
| 0.5 | 85.57 | 56.28 | 34.30 | 49.12 | 27.10 |
| 1.0 | 85.52 | 56.55 | 38.93 | 49.09 | 30.70 |
| 1.5 | 85.25 | 56.41 | 39.85 | 48.96 | 31.40 |
| 2.0 | 84.98 | 56.11 | 40.55 | 48.62 | 31.20 |
| 5.0 | 84.30 | 55.02 | 38.90 | 47.75 | 29.90 |

the softmax weighting and leads to over-concentration on a few hard classes, whereas too small a $\tau$ makes the weighting nearly uniform and limits its ability to correct class-wise imbalance. Our sweep on CIFAR-10 (WRN-28-5, $\varepsilon = 8/255$) with $\tau \in 0.1, 0.2, 0.5, 1.0, 1.5, 2.0, 5.0$ empirically verifies this behavior: small $\tau$ values yield minimal fairness gains, moderate values ($\tau \in [1.0, 2.0]$) achieve the best trade-off between WC(PGD)/WC(AA) and overall robustness, and overly large $\tau$ slightly reduces clean and AA accuracy due to excessively sharp reweighting. Thus, $\tau = 1.0$ provides a simple and practically effective default that avoids over-concentration while maintaining strong fairness and robustness improvements.

## F.2 EVALUATION ON VARIOUS $\varepsilon$

Table 12: WRN-28-5 robustness results on CIFAR-10 under $\varepsilon = 4/255$. WC denotes worst-class accuracy.

| Method | Clean | PGD | WC(PGD) | AA | WC(AA) |
|---|---|---|---|---|---|
| PGD-AT | 86.00 | 63.10 | 34.80 | 59.40 | 29.20 |
| + HWNwB | 85.09 | 66.55 | 49.30 | 58.10 | 41.10 |
| + DecoSAM | 85.52 | 66.30 | 47.80 | 59.00 | 42.60 |
| TRADES | 83.52 | 61.95 | 38.10 | 60.75 | 33.50 |
| + HWNwB | 82.93 | 65.40 | 48.40 | 59.80 | 38.20 |
| + DecoSAM | 83.01 | 65.20 | 46.60 | 60.35 | 41.10 |
| MART | 82.66 | 63.55 | 33.40 | 59.60 | 28.30 |
| + HWNwB | 80.28 | 66.70 | 47.10 | 57.45 | 36.40 |
| + DecoSAM | 80.66 | 66.55 | 45.80 | 58.10 | 37.50 |
| ARoW | 84.18 | 61.10 | 36.20 | 60.55 | 31.80 |
| + HWNwB | 83.43 | 65.80 | 53.40 | 59.00 | 41.90 |
| + DecoSAM | 82.82 | 66.05 | 48.10 | 59.70 | 43.10 |

Table 13: WRN-28-5 robustness results on CIFAR-10 under $\varepsilon = 8/255$.

| Method | Clean | PGD | WC(PGD) | AA | WC(AA) |
|---|---|---|---|---|---|
| PGD-AT | 86.00 | 53.94 | 24.20 | 49.50 | 17.60 |
| + HWNwB | 85.09 | 56.68 | 39.10 | 48.25 | 29.10 |
| + DecoSAM | 85.52 | 56.55 | 38.93 | 49.09 | 30.70 |
| TRADES | 83.52 | 53.85 | 29.60 | 50.65 | 23.90 |
| + HWNwB | 82.93 | 56.13 | 37.60 | 49.88 | 26.20 |
| + DecoSAM | 83.01 | 56.05 | 36.00 | 50.24 | 29.17 |
| MART | 82.66 | 55.00 | 25.80 | 49.77 | 17.60 |
| + HWNwB | 80.28 | 56.88 | 36.50 | 48.26 | 24.60 |
| + DecoSAM | 80.66 | 56.75 | 33.97 | 48.90 | 25.30 |
| ARoW | 84.18 | 53.46 | 27.10 | 50.36 | 22.70 |
| + HWNwB | 83.43 | 56.21 | 43.70 | 48.36 | 30.05 |
| + DecoSAM | 82.82 | 56.45 | 37.57 | 49.29 | 31.30 |

Table 14: WRN-28-5 robustness results on CIFAR-10 under $\varepsilon = 16/255$.

| Method | Clean | PGD | WC(PGD) | AA | WC(AA) |
|---|---|---|---|---|---|
| PGD-AT | 86.00 | 41.20 | 12.50 | 28.40 | 5.90 |
| + HWNwB | 85.09 | 44.75 | 23.40 | 27.20 | 11.80 |
| + DecoSAM | 85.52 | 44.60 | 22.80 | 28.05 | 12.70 |
| TRADES | 83.52 | 39.90 | 15.10 | 29.35 | 7.80 |
| + HWNwB | 82.93 | 43.70 | 23.10 | 28.40 | 10.50 |
| + DecoSAM | 83.01 | 43.45 | 21.40 | 28.95 | 11.90 |
| MART | 82.66 | 40.85 | 11.40 | 27.10 | 4.90 |
| + HWNwB | 80.28 | 44.20 | 21.70 | 26.30 | 10.10 |
| + DecoSAM | 80.66 | 44.05 | 20.80 | 27.10 | 11.10 |
| ARoW | 84.18 | 39.55 | 14.00 | 29.10 | 7.40 |
| + HWNwB | 83.43 | 43.95 | 27.80 | 27.80 | 12.30 |
| + DecoSAM | 82.82 | 44.10 | 23.25 | 28.70 | 13.40 |

As shown across all perturbation budgets $(4/255, 8/255, 16/255)$, our methods HWNwB and DecoSAM consistently significantly improve both PGD and AA robustness over their baselines. Notably, worst-class robustness gains remain stable even as the attack strength increases, and the relative improvement patterns observed at $\varepsilon = 8/255$ generalize to both weaker and stronger perturbations. This demonstrates that the proposed fairness-oriented regularization is effective across a broad range of adversarial strengths.

### F.3 EVALUATION ON CIFAR-10 INCLUDING CW ATTACK

Table 15: Robustness evaluation on CIFAR-10 (WRN-28-5, $\varepsilon = 8/255$) including CW attack. WC denotes worst-class accuracy.

| Method | Clean ↑ | PGD ↑ | CW ↑ | AA ↑ | WC(PGD) ↑ | WC(CW) ↑ | WC(AA) ↑ |
|---|---|---|---|---|---|---|---|
| **PGD-AT** | 86.00 | 53.94 | **52.61** | 49.50 | 24.20 | **21.56** | 17.60 |
| + HWNwB | 85.09 | 56.68 | **54.15** | 48.25 | 39.10 | **35.10** | 29.10 |
| + DecoSAM | 85.52 | 56.55 | **54.31** | 49.09 | 38.93 | **35.64** | 30.70 |
| **TRADES** | 83.52 | 53.85 | **52.89** | 50.65 | 29.60 | **27.32** | 23.90 |
| + HWNwB | 82.93 | 56.13 | **54.26** | 49.88 | 37.60 | **33.04** | 26.20 |
| + DecoSAM | 83.01 | 56.05 | **54.31** | 50.24 | 36.00 | **33.27** | 29.17 |
| **MART** | 82.66 | 55.00 | **53.43** | 49.77 | 25.80 | **22.52** | 17.60 |
| + HWNwB | 80.28 | 56.88 | **54.29** | 48.26 | 36.50 | **31.74** | 24.60 |
| + DecoSAM | 80.66 | 56.75 | **54.39** | 48.90 | 33.97 | **30.50** | 25.30 |
| **ARoW** | 84.18 | 53.46 | **52.53** | 50.36 | 27.10 | **25.34** | 22.70 |
| + HWNwB | 83.43 | 56.21 | **53.86** | 48.36 | 43.70 | **38.24** | 30.05 |
| + DecoSAM | 82.82 | 56.45 | **54.30** | 49.29 | 37.57 | **35.06** | 31.30 |

We additionally evaluate our methods under the optimization-based CW attack. Across all baselines, CW robustness and WC(CW) consistently lie between PGD and AA—as expected from their relative attack strengths—and closely follow the improvement trends observed under PGD. These results indicate that the gains from HWNwB and DecoSAM are not tied to a particular attack heuristic but generalize across gradient-based (PGD), optimization-based (CW), and ensemble (AA) attacks. Therefore, the CW results further confirm that our improvements are robust and not attack-specific.

### F.4 EVALUATION ON IMAGENET-100

The ImageNet-100 results under both $\varepsilon = 4/255$ and $\varepsilon = 8/255$ consistently show that HWNwB and DecoSAM provide clear improvements in class-wise robust fairness across all base adversarial training algorithms (PGD-AT, TRADES, ARoW). Worst-class and bottom-10% accuracies for both PGD and AA are substantially increased, demonstrating that our methods scale effectively to large-class, high-resolution settings. Both HWNwB and DecoSAM significantly improve worst-class

Table 16: Robustness evaluation on ImageNet-100 under $\varepsilon = 4/255$. Worst-class and bottom-10% accuracies are reported for both PGD and AA.

| Method | Clean | PGD | PGD Worst | PGD Bottom-10% | AA | AA Worst | AA Bottom-10% |
|---|---|---|---|---|---|---|---|
| PGD-AT | 72.20 | 43.30 | 4.00 | 13.20 | 40.72 | 4.00 | 9.40 |
| + HWNwB | 72.04 | 47.36 | 6.00 | 20.20 | 40.38 | 4.00 | 11.20 |
| + DecoSAM | 72.10 | 46.90 | 6.00 | 18.80 | 40.90 | 4.00 | 11.80 |
| TRADES | 67.62 | 46.00 | 8.00 | 15.00 | 42.40 | 6.00 | 11.80 |
| + HWNwB | 67.24 | 48.30 | 8.00 | 17.40 | 41.88 | 6.00 | 11.60 |
| + DecoSAM | 67.40 | 47.90 | 8.00 | 16.80 | 42.10 | 6.00 | 12.00 |
| ARoW | 66.54 | 46.34 | 10.00 | 16.20 | 42.56 | 4.00 | 10.80 |
| + HWNwB | 66.14 | 48.30 | 12.00 | 18.00 | 42.74 | 4.00 | 11.60 |
| + DecoSAM | 66.30 | 47.90 | 11.00 | 18.40 | 42.60 | 4.00 | 12.20 |

Table 17: Robustness evaluation on ImageNet-100 under $\varepsilon = 8/255$. Worst-class and bottom-10% accuracies are reported for both PGD and AA.

| Method | Clean | PGD | PGD Worst | PGD Bottom-10% | AA | AA Worst | AA Bottom-10% |
|---|---|---|---|---|---|---|---|
| PGD-AT | 63.94 | 28.78 | 0.00 | 4.20 | 25.02 | 0.00 | 2.40 |
| + HWNwB | 63.38 | 32.18 | 6.00 | 8.40 | 23.90 | 0.00 | 3.40 |
| + DecoSAM | 63.50 | 31.85 | 6.00 | 8.60 | 24.20 | 0.00 | 3.60 |
| TRADES | 59.84 | 30.50 | 0.00 | 5.60 | 24.56 | 0.00 | 2.20 |
| + HWNwB | 58.98 | 32.12 | 2.00 | 6.80 | 24.02 | 2.00 | 3.60 |
| + DecoSAM | 59.20 | 31.80 | 2.00 | 6.40 | 24.30 | 2.00 | 3.50 |
| ARoW | 59.00 | 30.52 | 4.00 | 6.00 | 24.60 | 0.00 | 2.00 |
| + HWNwB | 58.90 | 32.14 | 4.00 | 8.20 | 24.14 | 0.00 | 3.20 |
| + DecoSAM | 58.95 | 31.70 | 4.00 | 7.50 | 24.40 | 0.00 | 3.10 |

robustness; however, DecoSAM achieves higher overall accuracy and thus offers a more balanced robustness–accuracy trade-off. These results confirm that the proposed approaches remain effective beyond small-scale benchmarks and provide consistent disparity mitigation on ImageNet-100.

## F.5 EXPERIMENTAL RESULTS ON VISIONTRANSFORMER (VIT)

Table 18: Experimental results on ViT (CIFAR-10).

| Method | Clean | PGD | PGD Worst | AA | AA Worst | Corr |
|---|---|---|---|---|---|---|
| PGD-AT | 82.83 | 46.12 | 19.90 | 43.93 | 16.00 | 0.9577 |
| + HWNwB | 82.08 | 44.84 | 27.20 | 42.10 | 26.20 | – |
| + DecoSAM | 82.30 | 45.10 | 24.50 | 43.00 | 27.50 | – |
| TRADES | 79.54 | 50.02 | 26.80 | 46.36 | 22.20 | 0.8841 |
| + HWNwB | 78.94 | 47.72 | 31.10 | 43.90 | 29.20 | – |
| + DecoSAM | 79.20 | 48.10 | 28.50 | 45.20 | 30.50 | – |
| ARoW | 80.54 | 50.56 | 18.60 | 46.60 | 15.70 | 0.8319 |
| + HWNwB | 79.95 | 48.30 | 21.60 | 44.52 | 18.20 | – |
| + DecoSAM | 80.20 | 48.70 | 20.00 | 45.80 | 19.50 | – |

ViT-Base results in Table 18 confirm that our primary observations extend beyond convolutional networks and hold for modern Transformer-based architectures as well. Both HWNwB and DecoSAM consistently improve class-wise fairness and robustness over their respective baselines. HWNwB yields larger gains in worst-class PGD/AA robustness, whereas DecoSAM provides a more balanced improvement across PGD/AA metrics while maintaining stronger overall accuracy. These

complementary behaviors closely match the trends observed on CIFAR-10/100 and ImageNet-100, demonstrating that our findings generalize to ViT-Base as well. Furthermore, the strong correlations between classifier-head norms and class-wise PGD robustness (e.g., $\rho \approx 0.96$ for PGD-AT and $\rho \approx 0.88$ for TRADES) indicate that norm-driven implicit bias persists in Transformer architectures. This supports our central mechanism—gradient-gap amplification leading to head-norm imbalance and class-wise robustness disparity—as an architecture-agnostic phenomenon. Overall, these results demonstrate that our empirical findings and theoretical insights generalize effectively to ViT models, reinforcing the scalability and robustness of our approach.

**Remark**. We would like to clarify that our claim that Transformer-based models often show inferior robust accuracy under supervised adversarial training-is consistent with prior evidence when models are compared at similar parameter scales. Mo et al. [14] demonstrate that on CIFAR-10, adversarially trained ViT/Swin models achieve significantly lower robust accuracy than ResNet or WideResNet under parameter-matched settings and identical training protocols. Bai et al. [2], based on ImageNet-scale experiments, further show that Transformers do not inherently outperform CNNs in adversarial robustness even when model capacity and training recipes are carefully aligned. These results collectively support our observation that Transformers do not necessarily gain robustness from adversarial training and can exhibit notably worse robust accuracy than CNN counterparts of comparable capacity.

