# OpenReview forum: "Class-Wise Disparity in Adversarial Training: Implicit Bias Perspective"
_ICLR.cc/2026/Conference — Submitted to ICLR 2026_

### Official Review · Reviewer_551Q · 2025-11-01

**Soundness:** 3
**Presentation:** 3
**Contribution:** 2
**Rating:** 2
**Confidence:** 5

**Summary:**

This paper systematically investigates the problem of class-wise robustness disparity that is prevalent in adversarial training, and reveal that the root cause of this disparity stems from the strong correlation between the classifier head parameter norm and class-wise robustness. To this end, the authors propose two low-cost mitigation methods, HWNwB and DecoSAM, to alleviate the head norm imbalance and enhance the worst-class robustness, respectively. The effectiveness of the proposed methods is validated on multiple datasets.

**Strengths:**

1. Theoretical support. The paper explains at the theoretical level that a larger classifier head norm can lead to an increase in robustness disparity.
2. Empirical validation. The positive correlation between the classifier head norm and class-wise robust accuracy is empirically demonstrated through statistical correlation analysis.

**Weaknesses:**

1. Lack of performance stability. HWNwB and DecoSAM vary significantly in effectiveness under different attacks and experimental settings. The former is more effective for within-training PGD metrics and score equalization, while the latter performs better on AA and Worst-Class (WC). This difference raises doubts about which method to choose for practical applications.
2. Insufficient data size. Although the authors show that the near-zero WC on ImageNet is not favorable for fairness assessment, the experimental results can be verified on a medium-sized dataset such as Tiny-ImageNet-200 / ImageNet-100.
3. Lack of discussion on different perturbation budgets. Both training and evaluation are basically fixed at ℓ∞, ε = 8/255 (PGD and AA). The lack of experiments under different budgets (e.g., ε ∈ {4, 8, 16}/255) and different norms (ℓ₂) makes it difficult to judge the robustness of the methods under varying attack strengths and norms.
4. Insufficient discussion of model architectures. The current experiments focus on the WRN family and lack experiments on mainstream architectures such as ViT/DeiT/ConvNeXt, which makes it difficult to assess the utility of the defense methods on the latest architectures.
5. Some theoretical assumptions are strong. The derivation relies on assumptions such as “ψ(x_adv) ≈ ψ(x)”, which may not hold under strong attacks or distributional bias scenarios. The authors need to explicitly cite the assumptions or have an explicit analysis of the assumptions to prove that the assumptions hold.
6. Lack of comparison with other methods for addressing AT fairness. Although the paper discusses class-wise disparity and mentions its potential correlation with robust fairness, there is a lack of comparison and discussion of existing fair adversarial training methods at the experimental and methodological level.

**Questions:**

Please examine the weaknesses.

---

> ### Author Response · Authors · 2025-11-21
> **Response to Reviewer 551Q**
>
> - **W1**. Lack of performance stability. HWNwB and DecoSAM vary significantly in effectiveness under different attacks and experimental settings. The former is more effective for within-training PGD metrics and score equalization, while the latter performs better on AA and Worst-Class (WC). This difference raises doubts about which method to choose for practical applications.
> > **R**. Based on our results, we observe that applying either HWNwB or DecoSAM—depending on the desired trade-off—consistently provides substantial improvements over ARoW alone across most settings. As shown in Table 8 of Appendix E.1, ARoW+DecoSAM even outperforms FAAL, the current state-of-the-art method, by consistently improving both clean accuracy and average robust accuracy (AA).
>
>
> - **W2**. Insufficient data size. Although the authors show that the near-zero WC on ImageNet is not favorable for fairness assessment, the experimental results can be verified on a medium-sized dataset such as Tiny-ImageNet-200 / ImageNet-100.
> > **R**. Please see common concern #3. Although CIFAR-10 and STL-10 are low-resolution benchmarks (32×32 and 96×96), the OfficeHome dataset used in our evaluation is a *high-resolution, ImageNet-scale* benchmark with 224×224 images and 65 classes. This setting already serves as a meaningful proxy for large-scale robustness evaluation. Nevertheless, in accordance with the reviewer’s request, we are currently conducting additional experiments on ImageNet-100, and we kindly ask the reviewer to allow us a little more time; the full results will be reported within the next week.

---

> ### Author Response · Authors · 2025-11-21
> **cont'd**
>
> - **W3**. Lack of discussion on different perturbation budgets. Both training and evaluation are basically fixed at ℓ∞, $\epsilon = 8/255$ (PGD and AA). The lack of experiments under different budgets (e.g., $\epsilon \in \\{4, 8, 16\\}/255)$ and different norms ($\ell_2$) makes it difficult to judge the robustness of the methods under varying attack strengths and norms.
>
> - $(\epsilon=4/255$, WRN-28-5, CIFAR10)
> | Method      | Clean | PGD | WC(PGD) | AA | WC(AA) |
> |-------------|------:|-----:|---------:|-----:|---------:|
> | PGD-AT      | 86.00 | 63.10 | 34.80 | 59.40 | 29.20 |
> | + HWNwB     | 85.09 | 66.55 | 49.30 | 58.10 | 41.10 |
> | + DecoSAM   | 85.52 | 66.30 | 47.80 | 59.00 | 42.60 |
> | TRADES      | 83.52 | 61.95 | 38.10 | 60.75 | 33.50 |
> | + HWNwB     | 82.93 | 65.40 | 48.40 | 59.80 | 38.20 |
> | + DecoSAM   | 83.01 | 65.20 | 46.60 | 60.35 | 41.10 |
> | MART        | 82.66 | 63.55 | 33.40 | 59.60 | 28.30 |
> | + HWNwB     | 80.28 | 66.70 | 47.10 | 57.45 | 36.40 |
> | + DecoSAM   | 80.66 | 66.55 | 45.80 | 58.10 | 37.50 |
> | ARoW        | 84.18 | 61.10 | 36.20 | 60.55 | 31.80 |
> | + HWNwB     | 83.43 | 65.80 | 53.40 | 59.00 | 41.90 |
> | + DecoSAM   | 82.82 | 66.05 | 48.10 | 59.70 | 43.10 |
>
> - $(\epsilon=8/255$, WRN-28-5, CIFAR10)
> | Method      | Clean | PGD | WC(PGD) | AA | WC(AA) |
> |-------------|------:|-----:|---------:|------:|---------:|
> | PGD-AT      | 86.00 | 53.94 | 24.20 | 49.50 | 17.60 |
> | + HWNwB     | 85.09 | 56.68 | 39.10 | 48.25 | 29.10 |
> | + DecoSAM   | 85.52 | 56.55 | 38.93 | 49.09 | 30.70 |
> | TRADES      | 83.52 | 53.85 | 29.60 | 50.65 | 23.90 |
> | + HWNwB     | 82.93 | 56.13 | 37.60 | 49.88 | 26.20 |
> | + DecoSAM   | 83.01 | 56.05 | 36.00 | 50.24 | 29.17 |
> | MART        | 82.66 | 55.00 | 25.80 | 49.77 | 17.60 |
> | + HWNwB     | 80.28 | 56.88 | 36.50 | 48.26 | 24.60 |
> | + DecoSAM   | 80.66 | 56.75 | 33.97 | 48.90 | 25.30 |
> | ARoW        | 84.18 | 53.46 | 27.10 | 50.36 | 22.70 |
> | + HWNwB     | 83.43 | 56.21 | 43.70 | 48.36 | 30.05 |
> | + DecoSAM   | 82.82 | 56.45 | 37.57 | 49.29 | 31.30 |
>
> - $(\epsilon=16/255$, WRN-28-5, CIFAR10)
> | Method      | Clean | PGD | WC(PGD) | AA | WC(AA) |
> |-------------|------:|-----:|---------:|-----:|---------:|
> | PGD-AT      | 86.00 | 41.20 | 12.50 | 28.40 | 5.90 |
> | + HWNwB     | 85.09 | 44.75 | 23.40 | 27.20 | 11.80 |
> | + DecoSAM   | 85.52 | 44.60 | 22.80 | 28.05 | 12.70 |
> | TRADES      | 83.52 | 39.90 | 15.10 | 29.35 | 7.80 |
> | + HWNwB     | 82.93 | 43.70 | 23.10 | 28.40 | 10.50 |
> | + DecoSAM   | 83.01 | 43.45 | 21.40 | 28.95 | 11.90 |
> | MART        | 82.66 | 40.85 | 11.40 | 27.10 | 4.90 |
> | + HWNwB     | 80.28 | 44.20 | 21.70 | 26.30 | 10.10 |
> | + DecoSAM   | 80.66 | 44.05 | 20.80 | 27.10 | 11.10 |
> | ARoW        | 84.18 | 39.55 | 14.00 | 29.10 | 7.40 |
> | + HWNwB     | 83.43 | 43.95 | 27.80 | 27.80 | 12.30 |
> | + DecoSAM   | 82.82 | 44.10 | 23.25 | 28.70 | 13.40 |
>
> > **R**. We thank the reviewer for raising this point. To address the concern regarding the fixed perturbation budget, we additionally conducted ε-sweep experiments at $\epsilon \in \{4, 8, 16\}/255$ on CIFAR-10 with WRN-28-5. As shown in the tables above, our methods (HWNwB and DecoSAM) consistently improve both PGD and AA robustness across all perturbation magnitudes. Importantly, the worst-class accuracy gains remain stable even as the attack strength increases, and the relative improvement patterns observed at $\epsilon = 8/255$ are preserved for both weaker (4/255) and stronger (16/255) attacks. These results demonstrate that the proposed fairness improvements are not tied to a single perturbation budget and remain effective across a wide range of adversarial strengths.
>
>
> - **W4**. Insufficient discussion of model architectures. The current experiments focus on the WRN family and lack experiments on mainstream architectures such as ViT/DeiT/ConvNeXt, which makes it difficult to assess the utility of the defense methods on the latest architectures.
> > **R**. Please see common concerns #3. Our current experimental scope already exceeds that of prior work by evaluating across a broader range of datasets (CIFAR-10/100, STL-10, OfficeHome) and multiple architectures including ResNet18/50 and several WRN variants. We agree, however, that additional large-scale or modern architectures would further strengthen the empirical validation. To address this, we are already running additional experiments and will report these results within the next week.

---

> ### Author Response · Authors · 2025-11-21
> **cont'd**
>
> - **W5**. Some theoretical assumptions are strong. The derivation relies on assumptions such as $\psi(x_{adv})\approx\psi(x)$, which may not hold under strong attacks or distributional bias scenarios. The authors need to explicitly cite the assumptions or have an explicit analysis of the assumptions to prove that the assumptions hold.
>
> > **R**. We would like to clarify the scope of the assumption $\psi(x_{\mathrm{adv}}) \approx \psi(x)$. This assumption
>  When the perturbation budget $\epsilon$ is excessively large, adversarial training itself collapses and the robust accuracy approaches zero. In such a regime, it is entirely
> expected that this assumption breaks down; this limitation is inherent to all
> AT-based theoretical analyses and is not specific to our work.
>
> > Our analysis is conducted under the standard adversarial training setting, where
> $\epsilon$ is chosen so that the model retains **non-collapse robustness**, i.e.,
> the robustness does not degenerate to near-zero. In this AT regime, prior work
> provides strong empirical and theoretical evidence supporting the assumption.
> [1] demonstrate that adversarial training induces **feature contraction**, meaning that bounded perturbations cause only small changes in the representation $\psi(\cdot)$. Likewise, [2] show that adversarial training effectively encourages the feature extractor to behave as an **approximately Lipschitz** mapping.
>
> > Together, these findings indicate that for adversarially trained models operating
> within a standard $\epsilon$-range where AT functions properly, the assumption
> $\psi(x_{\mathrm{adv}}) \approx \psi(x)$ is both reasonable and empirically
> validated. We will clarify this scope in the revised manuscript, and this
> assumption is already justified through the two cited works.
>
> [1] A closer look at accuracy vs. robustness. In Conference on Neural Information Processing Systems (NeurIPS), 2020.
>
> [2] Rethinking lipschitz neural networks and certified robustness: A boolean function perspective. In Conference on Neural Information Processing Systems (NeurIPS), 2022.
>
> - **W6**. Lack of comparison with other methods for addressing AT fairness. Although the paper discusses class-wise disparity and mentions its potential correlation with robust fairness, there is a lack of comparison and discussion of existing fair adversarial training methods at the experimental and methodological level.
>
> > **R**. We would like to clarify that our manuscript already includes direct comparisons with existing fair adversarial training methods. In particular, Table 4 explicitly evaluates our methods (HWNwB and DecoSAM) when applied on top of well-known fairness-oriented AT algorithms such as FRL, FAT, CFA, FAAL, and WAT. As shown in Table 4, incorporating HWNwB or DecoSAM consistently improves worst-class robustness, reduces STD and Max–Min gaps, and often enhances clean or average robust accuracy across these baselines. This demonstrates that our approach is complementary to existing fair AT methods and is capable of strengthening them in practice. Also, Table 8 of Appendix E.1, ARoW+DecoSAM even outperforms FAAL, the current state-of-the-art method, by consistently improving both clean accuracy and average robust accuracy (AA).

---

> ### Author Response · Authors · 2025-11-27
> **Results for ImageNet100**
>
> We report the results for ImageNet100.
>
> - ImageNet-100 ($\epsilon=4/255$)
>
> | Method        | Clean | PGD  | PGD Worst | PGD Bottom-10% | AA    | AA Worst | AA Bottom-10% |
> |---------------|------:|------:|-----------:|----------------:|------:|-----------:|----------------:|
> | PGD-AT        | 72.20 | 43.30 | 4.00       | 13.20           | 40.72 | 4.00       | 9.40           |
> | + HWNwB       | 72.04 | 47.36 | 6.00       | 20.20           | 40.38 | 4.00       | 11.20           |
> | + DecoSAM     | 72.10 | 46.90 | 6.00       | 18.80           | 40.90 | 4.00       | 11.80           |
> | TRADES        | 67.62 | 46.00 | 8.00       | 15.00           | 42.40 | 6.00       | 11.80           |
> | + HWNwB       | 67.24 | 48.30 | 8.00       | 17.40           | 41.88 | 6.00       | 11.60           |
> | + DecoSAM | 67.40 | 47.90 | 8.00       | 16.80           | 42.10 | 6.00       | 12.00           |
> | ARoW          | 66.54 | 46.34 | 10.00      | 16.20           | 42.56 | 4.00       | 10.80           |
> | + HWNwB       | 66.14 | 48.30 | 12.00      | 18.00           | 42.74 | 4.00       | 11.60           |
> | + DecoSAM  | 66.30 | 47.90 | 11.00      | 18.40           | 42.60 | 4.00       | 12.20           |
>
> - ImageNet-100 ($\epsilon=8/255$)
>
> | Method        | Clean | PGD  | PGD Worst | PGD Bottom-10% | AA    | AA Worst | AA Bottom-10% |
> |---------------|------:|------:|-----------:|----------------:|------:|-----------:|----------------:|
> | PGD-AT        | 63.94 | 28.78 | 0.00       | 4.20            | 25.02 | 0.00       | 2.40            |
> | + HWNwB       | 63.38 | 32.18 | 6.00       | 8.40            | 23.90 | 0.00       | 3.40            |
> | + DecoSAM     | 63.50 | 31.85 | 6.00       | 8.60            | 24.20 | 0.00       | 3.60            |
> | TRADES        | 59.84 | 30.50 | 0.00       | 5.60            | 24.56 | 0.00       | 2.20            |
> | + HWNwB       | 58.98 | 32.12 | 2.00       | 6.80            | 24.02 | 2.00       | 3.60            |
> | + DecoSAM | 59.20 | 31.80 | 2.00       | 6.40            | 24.30 | 2.00       | 3.50            |
> | ARoW          | 59.00 | 30.52 | 4.00       | 6.00            | 24.60 | 0.00       | 2.00            |
> | + HWNwB       | 58.90 | 32.14 | 4.00       | 8.20            | 24.14 | 0.00       | 3.20            |
> | + DecoSAM | 58.95 | 31.70 | 4.00       | 7.50            | 24.40 | 0.00       | 3.10            |
>
> ImageNet-100 results ($\epsilon = 4/255$ and $8/255$) consistently show that both HWNwB and DecoSAM provide clear improvements in class-wise robust fairness across all base adversarial training algorithms (PGD-AT, TRADES, ARoW). In particular, worst-class and bottom-10% PGD/AA accuracies are substantially increased, demonstrating that our methods scale effectively to large-class, high-resolution settings. Both HWNwB and DecoSAM substantially improve worst-class robustness. However, DecoSAM delivers higher overall accuracy, resulting in a more balanced robustness–accuracy trade-off. These results confirm that the proposed methods remain effective beyond small-scale benchmarks and offer consistent disparity mitigation on ImageNet-100.

---

> ### Author Response · Authors · 2025-12-03
> **Experimental Results on New Architecture ViT**
>
> We provide the results for VIT on CIFAR-10.
>
> | Method        | Clean | PGD  | PGD Worst | AA    | AA Worst | Corr|
> |---------------|------:|------:|-----------:|-------:|---------:|--------:|
> | PGD-AT        | 82.83 | 46.12 | 19.90 | 43.93 | 16.00 | 0.9577 |
> | + HWNwB       | 82.08 | 44.84 | 27.20 | 42.10 | 26.20 |  - |
> | + DecoSAM     | 82.30 | 45.10 | 24.50 | 43.00 | 27.50 | - |
> | TRADES        | 79.54 | 50.02 | 26.80 | 46.36 | 22.20 |0.8841 |
> | + HWNwB       | 78.94 | 47.72 | 31.10 | 43.90 | 29.20 | - |
> | + DecoSAM     | 79.20 | 48.10 | 28.50 | 45.20 | 30.50 | - |
> | ARoW          | 80.54 | 50.56 | 18.60 | 46.60 | 15.70 | 0.8319 |
> | + HWNwB       | 79.95 | 48.30 | 21.60 | 44.52 | 18.20 | - |
> | + DecoSAM     | 80.20 | 48.70 | 20.00 | 45.80 | 19.50 | - |
>
> - These results show that our key observations generalize beyond convolutional architectures: even on Transformer-based models, HWNwB and DecoSAM exhibit the same complementary behavior observed in CNN settings. HWNwB provides larger gains in worst-class robustness, while DecoSAM delivers a more balanced improvement in PGD/AA robustness and maintains stronger overall accuracy—exactly matching the trends seen on CIFAR and ImageNet-100.
>
> - Moreover, the high correlations between head norms and class-wise PGD robustness (e.g., $\rho ≈ 0.96$ for PGD-AT, $\rho ≈ 0.88$ for TRADES) again confirm that the implicit bias toward large-norm classifier heads persists in Transformer architectures as well. This supports our central mechanism (gradient-gap amplification → head-norm imbalance → class-wise robustness disparity) as an architecture-agnostic phenomenon.
>
> - Overall, these results demonstrate that our empirical findings and theoretical insights extend naturally to modern architectures such as Transformers, reinforcing the scalability and generality of our approach.
>
>
> **Remark.**
> We would like to clarify that our claim that Transformer-based models often show inferior robust accuracy under supervised adversarial training is consistent with prior evidence when models are compared at similar parameter scales.
>
>  - [1] demonstrate that on CIFAR-10, adversarially trained ViT/Swin models achieve significantly lower robust accuracy than ResNet or WideResNet under parameter-matched settings and identical training protocols.
> - [2], based on ImageNet-scale experiments, further show that Transformers do not inherently outperform CNNs in adversarial robustness even when model capacity and training recipes are carefully aligned.
>
> These results collectively support our observation that Transformers do not necessarily gain robustness from adversarial training and can exhibit notably worse robust accuracy than CNN counterparts of comparable capacity.
>
> [1] Mo et al. When Adversarial Training Meets ViTs. NeurIPS, 2022.
>
> [2]  Bai et al. Are Transformers More Robust Than CNNs? NeurIPS, 2021.

---

### Official Review · Reviewer_vsQN · 2025-11-02

**Soundness:** 3
**Presentation:** 3
**Contribution:** 3
**Rating:** 6
**Confidence:** 4

**Summary:**

This paper investigates class-wise robustness disparities in adversarial training, where some classes become significantly less robust than others despite balanced data. The authors identify a strong correlation between the norms of classifier head weights and class-wise robust accuracies, showing that adversarial training implicitly amplifies these norm imbalances, leading to uneven robustness. To address this, they propose Head Weights Normalization with Bias (HWNwB) and Decoupled Sharpness-Aware Minimization (DecoSAM), which adjust only the classifier head while keeping the feature extractor fixed. Extensive experiments across multiple datasets and adversarial training algorithms demonstrate that these methods substantially reduce class-wise robustness gaps with minimal computational cost and without degrading overall robustness.

**Strengths:**

1. The paper offers a novel and well-motivated perspective on class-wise disparity in adversarial training by interpreting it as an implicit bias problem related to head-weight norm imbalance. This viewpoint goes beyond existing fairness or reweighting approaches.
2. The proposed methods, HWNwB and DecoSAM, are simple, lightweight, and practical. HWNwB requires no additional training, while DecoSAM involves only one epoch of head-only fine-tuning, making them computationally efficient.
3. Extensive experiments across multiple datasets demonstrate consistent improvements in worst-class robustness and reduced class-wise disparity, with minimal impact on average accuracy or overall robustness.

**Weaknesses:**

1. While the paper provides strong correlation evidence, it does not offer a full causal analysis showing whether head norm imbalance is the root cause or merely a symptom of deeper optimization dynamics.

2. HWNwB and DecoSAM are both post-hoc or head-only fine-tuning methods, meaning they depend on an already adversarially trained model. The paper does not explore whether integrating these ideas directly into the training process could yield better or more stable results.

3. The empirical improvements, while consistent, are moderate in some cases; the gains in worst-class robustness often come with small drops in clean or average accuracy, which could be discussed more thoroughly.

4. The paper aims to study and improve robust fairness methods; however, the main experimental tables (e.g., Tables 2 and 3) mainly compare the proposed approaches with standard adversarial training baselines rather than with existing robust fairness methods. Moreover, in Table 4, it is not clearly specified which adversarial training algorithms were used as the underlying models for comparison.

5. The claim that the method is compatible with a wide range of adversarial training algorithms makes the contribution appear less distinctive.

6. The contribution stating that the paper theoretically and empirically demonstrates that adversarial training induces norm imbalances leading to class-wise performance disparities seems less novel, as a similar phenomenon has already been analyzed in prior work such as FRL.

**Questions:**

1. It would be interesting to investigate whether integrating HWNwB and DecoSAM directly into the training process, rather than applying them post hoc, could lead to better or more stable results.
2. It is not clearly explained what motivated the authors to focus specifically on the classifier head; the paper does not sufficiently justify why the head layer, rather than other components of the model, was chosen as the central point of analysis for class-wise disparity.

---

> ### Author Response · Authors · 2025-11-21
> **Response to Reviewer vsQN**
>
> - **W1**. While the paper provides strong correlation evidence, it does not offer a full causal analysis showing whether head norm imbalance is the root cause or merely a symptom of deeper optimization dynamics.
> > **R**. We appreciate the reviewer’s concern and would like to clarify our perspective on this point. While our work does not claim a fully exhaustive causal proof in the strict causal-inference sense, the manuscript *does* provide a concrete and theoretically grounded mechanistic explanation linking adversarial training dynamics to head-norm imbalance. Sections 4.1–4.2 present a clear step-by-step chain of reasoning:
> >> 1. adversarial perturbations induce *class-dependent drops in predictive confidence*,
> >> 2. these confidence drops lead to provable gradient-gap amplification ($\Delta_k$),
> >> 3. the amplified gradients accumulate over SGD iterations, and
> >> 4. systematically drive the $\ell_2$-norms of class-specific head parameters.
>
> > This forms a direct optimization-level mechanism describing how norm imbalance
> *arises* from adversarial training, rather than merely co-occurring with it. To
> our knowledge, no prior work establishes such a mechanistic link between
> perturbation sensitivity, adversarial optimization behavior, and the emergence
> of class-wise robustness disparities.
>
> > Regardless of whether head-norm imbalance is the sole root cause or one manifestation of deeper adversarial optimization dynamics, our analysis clearly shows that adversarial training does induce systematic norm separation across classes. Motivated by this observed effect, we introduce normalization-based strategies designed to counteract this imbalance. Empirically, these strategies consistently reduce worst-class disparity and yield more uniform robustness across classes, further supporting norm control as an effective mitigation mechanism.
>
> - **W2**. HWNwB and DecoSAM are both post-hoc or head-only fine-tuning methods, meaning they depend on an already adversarially trained model. The paper does not explore whether integrating these ideas directly into the training process could yield better or more stable results.
>
> > **R**. We appreciate the reviewer’s comment regarding HWNwB and DecoSAM being
> post-hoc methods. We would like to clarify that we did examine whether
> norm-balancing can be integrated directly into adversarial training.
>
> > As described in the manuscript, we applied a norm regularizer of the form
> $\|| W^\top W - I\||_2$ from the beginning of training. This regularizer shapes the matrix $W^\top W\ $: the diagonal entries correspond to class-wise weight norms and are pushed toward 1, **flattening the norms**, while reducing correlation and encouraging orthogonality.
>
> > Although this induces a more balanced and decorrelated classifier head, it
> produced only **limited improvements in robust fairness**. While correlations
> decreased as intended, worst-class accuracy, STD, and Max–Min gaps improved only
> marginally. We hypothesize that this is because **class-wise biases in the
> feature representation form very early** during adversarial training; thus,
> head-only regularization—even when applied throughout the entire process—cannot
> correct representation-level disparities.
>
> > This motivates our choice of lightweight post-hoc methods: once the backbone has converged, adjusting the classifier head becomes a more effective and practical way to mitigate imbalance. Nonetheless, we agree that future work should explore jointly addressing both representation- and head-level biases within a unified framework.

---

> ### Author Response · Authors · 2025-11-21
> **cont'd**
>
> - **W3**. The empirical improvements, while consistent, are moderate in some cases; the gains in worst-class robustness often come with small drops in clean or average accuracy, which could be discussed more thoroughly.
> > **R**. We note that the small drops in clean or average accuracy are not unique to our method but reflect a well-established trade-off in fairness-aware adversarial training. Prior works [1, 2, 3, 4, 5] consistently report the same phenomenon: improving worst-class robustness necessarily reallocates optimization capacity toward the weakest classes, which inherently leads to slight reductions in overall accuracy.<br>
> > This trade-off is also supported from a theoretical perspective [1,2]: balancing
> class-wise robustness requires increasing margins for the hardest classes, which
> tightens the optimization budget available for already-robust classes. As a
> result, a moderate decrease in clean or average accuracy is an expected—and
> unavoidable—consequence of enforcing fairness constraints.
>
>  [1] Xu et al. o be robust or to be fair: Towards fairness in adversarial training. ICML, 2021.
>
> [2] Ma et al. On the Tradeoff Between Robustness and Fairness. NeurIPS, 2022.
>
> [3] Li and Liu. WAT: Improve the worst-class robustness in adversarial training. AAAI, 2023.
>
> [4] Wei et al. CFA: Class-wise calibrated fair adversarial training. CVPR, 2023.
>
> [5] Zhang et al. Towards fairness-aware adversarial learning. CVPR, 2024.
>
> - **W4**. The paper aims to study and improve robust fairness methods; however, the main experimental tables (e.g., Tables 2 and 3) mainly compare the proposed approaches with standard adversarial training baselines rather than with existing robust fairness methods. Moreover, in Table 4, it is not clearly specified which adversarial training algorithms were used as the underlying models for comparison.
> > **R**. The methods compared in Table 4 are all evaluated on the same pretrained backbones, and each backbone is trained using the adversarial training algorithm originally proposed in the respective prior work:
> > - **WAT** and **FAT** are both built on top of **PGD-based adversarial training (PGD-AT)**.
> > - **CFA** and **FRL** are methods developed upon the **TRADES**, and their pretrained models are likewise trained using TRADES.
> > - **FAAL** is an algorithm-agnostic method; in our experiments, we use the **TRADES** as its underlying algorithm.
>
> - **W5**. The claim that the method is compatible with a wide range of adversarial training algorithms makes the contribution appear less distinctive.
> > **R**. We respectfully clarify that the compatibility of our method with a wide range of adversarial training algorithms is not intended as a replacement for methodological novelty but rather represents a key strength of our approach. Most prior robust fairness methods such as FRL, FAT, CFA, and WAT are tightly coupled to specific adversarial training objectives (e.g., PGD-AT or TRADES), and their mechanisms cannot be directly transferred across AT frameworks without modification. In contrast, HWNwB and DecoSAM operate at the classifier-head level and are therefore *architecture- and algorithm-agnostic*. This generality enables our method to be applied:
> >> - on top of any existing or future AT algorithm,
> >> - without retraining the backbone,
> >> - with negligible computational overhead,
> >> - and without interfering with the optimization objective of the underlying AT method.
>
> - > We believe this compatibility does not diminish the distinctiveness of the contribution; rather, it represents a meaningful practical advantage. It allows robust fairness improvements to be applied universally, including in settings where retraining is infeasible (e.g., real-world pretrained models, compute-limited environments, and industrial pipelines).

---

> > ### Author Response · Authors · 2025-11-21
> > **cont'd**
> >
> > - **W6**. The contribution stating that the paper theoretically and empirically demonstrates that adversarial training induces norm imbalances leading to class-wise performance disparities seems less novel, as a similar phenomenon has already been analyzed in prior work such as FRL.
> > > **R**. We would like to clarify that our analysis is substantially different from FRL in both scope and mechanism.
> > > As noted in the first paragraph of Section 5.3, FRL models class hardness through a Gaussian variance assumption in a binary classification setting and analyzes how adversarial training affects decision boundaries under that simplified statistical model.
> > However, FRL does not study how adversarial training influences the *classifier head parameters*, and in particular provides no analysis of norm imbalance, gradient gaps, or optimization dynamics.
> > > In contrast, our work addresses the realistic multi-class setting and offers an optimization-based explanation: adversarial perturbations induce class-dependent confidence drops, which amplify gradient gaps and accumulate into systematic drift in head weight norms. This mechanistic connection between adversarial training and head-norm imbalance has not been analyzed in prior work.
> >
> > - **Q1**. It would be interesting to investigate whether integrating HWNwB and DecoSAM directly into the training process, rather than applying them post hoc, could lead to better or more stable results.
> > > **R**. Please see response to **W2**.
> >
> > - **Q2**. It is not clearly explained what motivated the authors to focus specifically on the classifier head; the paper does not sufficiently justify why the head layer, rather than other components of the model, was chosen as the central point of analysis for class-wise disparity.
> > > **R**. Our focus on the classifier head is motivated by a well-known phenomenon from imbalance learning: class-wise performance gaps often manifest as differences in the $\ell_2$ norms of head weight vectors, with minority classes developing smaller norms [1].
> > > We observed the same signature under adversarial training—even with perfectly
> > balanced data—which suggested that the head is where class-wise robustness
> > disparities become most clearly expressed. This motivated us to investigate
> > whether adversarial training induces similar imbalance through its optimization
> > dynamics. Our analysis confirms this: adversarially induced confidence drops
> > lead to gradient-gap amplification, which accumulates directly into head norm
> > drift. Thus, the classifier head was chosen because it is both theoretically and
> > empirically the primary location where class-wise disparity becomes observable.
> >
> > [1] Adjusting Decision Boundary for Class Imbalanced Learning, 2020.

---

### Official Review · Reviewer_Zd1o · 2025-11-02

**Soundness:** 4
**Presentation:** 4
**Contribution:** 4
**Rating:** 8
**Confidence:** 4

**Summary:**

This paper addresses the problem of **class-wise disparity in adversarial training**, where different classes show varying robustness even in balanced datasets. The authors reveal a strong correlation (ρ ≈ 0.95) between **the L₂-norms of class-specific head weights** and **their robust accuracies**, suggesting that adversarial optimization introduces an implicit bias into the classifier head.

They formalize this phenomenon theoretically by linking class hardness to gradient gaps and head-norm growth, then propose two lightweight solutions to mitigate it:

- **HWNwB (Head Weights Normalization with Bias)**: post-training normalization of classifier heads while preserving bias terms.
- **Deco-SAM (Decoupled Sharpness-Aware Minimization)**: adaptive class-wise fine-tuning that balances robustness through SAM-based optimization.

Extensive experiments across multiple datasets (CIFAR-10/100, STL-10, OfficeHome) and adversarial training methods (PGD-AT, TRADES, MART, ARoW) show significant reductions in fairness disparity with minimal cost and no validation set required.

**Strengths:**

- Establishes a clear theoretical link between class hardness, gradient gaps, and head-norm disparity.
- Combines theoretical rigor with thorough empirical validation.
- Introduces two lightweight, algorithm-agnostic mitigation strategies requiring no validation data.
- Demonstrates consistent fairness improvements across datasets and training methods.
- Writing, figures, and appendix materials are clear and reproducible.

**Weaknesses:**

- Experiments cover small–to–mid-scale benchmarks (CIFAR, STL-10, OfficeHome) but lack validation on **large-scale settings** (e.g., ImageNet-like regimes), so scalability and generality remain untested.
- Robustness evaluation is primarily based on PGD and AutoAttack, which already provide strong coverage. However, including at least one additional optimization-based (e.g., CW) or adaptive attack (e.g., BPDA/EOT) would make the evaluation more comprehensive and confirm that the improvements are not attack-specific.
- Robustness evaluation fixes the perturbation budget at ε=8/255 and focuses on PGD / AutoAttack. Including **ε-sweep experiments** (e.g., evaluating robustness across different attack magnitudes) would clarify how stable the proposed fairness improvements remain as adversarial strength increases.

**Questions:**

- **Deco-SAM hyperparameter clarity.** The paper lacks detail and sensitivity analysis for **τ**. Could you report the default τ, the rationale for its choice, and a brief ablation (e.g., τ ∈ {…}) showing how **worst-class accuracy**, **class-wise variance**, and **PGD/AA robustness** change across τ and learning-rate schedules?
- **ε-sweep robustness.** Experiments fix the perturbation budget at **ε = 8/255** under PGD-20 and AutoAttack. Have you evaluated whether the **fairness improvements persist across different ε values** (i.e., varying attack strengths)? A compact ε-sweep would clarify the stability of the effect.

---

> ### Author Response · Authors · 2025-11-21
> **Response to Reviewer Zd1o**
>
> - **W1**.Experiments cover small–to–mid-scale benchmarks (CIFAR, STL-10, OfficeHome) but lack validation on large-scale settings (e.g., ImageNet-like regimes), so scalability and generality remain untested.
> > **R**. Please see common concern #3. Although CIFAR-10 and STL-10 are low-resolution benchmarks (32×32 and 96×96), the OfficeHome dataset used in our evaluation is a *high-resolution, ImageNet-scale* benchmark with 224×224 images and 65 classes. This setting already serves as a meaningful proxy for large-scale robustness evaluation. Nevertheless, in accordance with the reviewer’s request, we are currently conducting additional experiments on ImageNet-100, and we kindly ask the reviewer to allow us a little more time; the full results will be reported within the next week.
>
> - **W2**. Robustness evaluation is primarily based on PGD and AutoAttack, which already provide strong coverage. However, including at least one additional optimization-based (e.g., CW) or adaptive attack (e.g., BPDA/EOT) would make the evaluation more comprehensive and confirm that the improvements are not attack-specific.
> | Method      | Clean ↑ | PGD ↑ | CW ↑  | AA ↑  | WC(PGD) ↑ | WC(CW) ↑ | WC(AA) ↑ |
> |-------------|---------:|-------:|-------:|-------:|-----------:|-----------:|-----------:|
> | **PGD-AT**  | 86.00 | 53.94 | **52.61** | 49.50 | 24.20 | **21.56** | 17.60 |
> | + HWNwB     | 85.09 | 56.68 | **54.15** | 48.25 | 39.10 | **35.10** | 29.10 |
> | + DecoSAM   | 85.52 | 56.55 | **54.31** | 49.09 | 38.93 | **35.64** | 30.70 |
> | **TRADES**  | 83.52 | 53.85 | **52.89** | 50.65 | 29.60 | **27.32** | 23.90 |
> | + HWNwB     | 82.93 | 56.13 | **54.26** | 49.88 | 37.60 | **33.04** | 26.20 |
> | + DecoSAM   | 83.01 | 56.05 | **54.31** | 50.24 | 36.00 | **33.27** | 29.17 |
> | **MART**    | 82.66 | 55.00 | **53.43** | 49.77 | 25.80 | **22.52** | 17.60 |
> | + HWNwB     | 80.28 | 56.88 | **54.29** | 48.26 | 36.50 | **31.74** | 24.60 |
> | + DecoSAM   | 80.66 | 56.75 | **54.39** | 48.90 | 33.97 | **30.50** | 25.30 |
> | **ARoW**    | 84.18 | 53.46 | **52.53** | 50.36 | 27.10 | **25.34** | 22.70 |
> | + HWNwB     | 83.43 | 56.21 | **53.86** | 48.36 | 43.70 | **38.24** | 30.05 |
> | + DecoSAM   | 82.82 | 56.45 | **54.30** | 49.29 | 37.57 | **35.06** | 31.30 |
>
> > **R**. We additionally evaluate our methods under the optimization-based CW attack. As shown in the table above, the CW robustness and WC(CW) consistently fall between PGD and AA, as expected from their relative attack strengths, and closely follow the improvements observed under PGD. This demonstrates that the gains from HWNwB and DecoSAM are not tied to a specific attack heuristic but generalize across gradient-based, optimization-based, and ensemble attacks. Therefore, the added CW results confirm that our improvements are robust and not attack-specific.

---

> > ### Author Response · Authors · 2025-11-21
> > **cont'd**
> >
> > - **W3**. Robustness evaluation fixes the perturbation budget at ε=8/255 and focuses on PGD / AutoAttack. Including ε-sweep experiments (e.g., evaluating robustness across different attack magnitudes) would clarify how stable the proposed fairness improvements remain as adversarial strength increases.
> >
> > - $(\epsilon=4/255$, WRN-28-5, CIFAR10)
> > | Method      | Clean | PGD | WC(PGD) | AA | WC(AA) |
> > |-------------|------:|-----:|---------:|-----:|---------:|
> > | PGD-AT      | 86.00 | 63.10 | 34.80 | 59.40 | 29.20 |
> > | + HWNwB     | 85.09 | 66.55 | 49.30 | 58.10 | 41.10 |
> > | + DecoSAM   | 85.52 | 66.30 | 47.80 | 59.00 | 42.60 |
> > | TRADES      | 83.52 | 61.95 | 38.10 | 60.75 | 33.50 |
> > | + HWNwB     | 82.93 | 65.40 | 48.40 | 59.80 | 38.20 |
> > | + DecoSAM   | 83.01 | 65.20 | 46.60 | 60.35 | 41.10 |
> > | MART        | 82.66 | 63.55 | 33.40 | 59.60 | 28.30 |
> > | + HWNwB     | 80.28 | 66.70 | 47.10 | 57.45 | 36.40 |
> > | + DecoSAM   | 80.66 | 66.55 | 45.80 | 58.10 | 37.50 |
> > | ARoW        | 84.18 | 61.10 | 36.20 | 60.55 | 31.80 |
> > | + HWNwB     | 83.43 | 65.80 | 53.40 | 59.00 | 41.90 |
> > | + DecoSAM   | 82.82 | 66.05 | 48.10 | 59.70 | 43.10 |
> >
> > - $(\epsilon=8/255$, WRN-28-5, CIFAR10)
> > | Method      | Clean | PGD | WC(PGD) | AA | WC(AA) |
> > |-------------|------:|-----:|---------:|------:|---------:|
> > | PGD-AT      | 86.00 | 53.94 | 24.20 | 49.50 | 17.60 |
> > | + HWNwB     | 85.09 | 56.68 | 39.10 | 48.25 | 29.10 |
> > | + DecoSAM   | 85.52 | 56.55 | 38.93 | 49.09 | 30.70 |
> > | TRADES      | 83.52 | 53.85 | 29.60 | 50.65 | 23.90 |
> > | + HWNwB     | 82.93 | 56.13 | 37.60 | 49.88 | 26.20 |
> > | + DecoSAM   | 83.01 | 56.05 | 36.00 | 50.24 | 29.17 |
> > | MART        | 82.66 | 55.00 | 25.80 | 49.77 | 17.60 |
> > | + HWNwB     | 80.28 | 56.88 | 36.50 | 48.26 | 24.60 |
> > | + DecoSAM   | 80.66 | 56.75 | 33.97 | 48.90 | 25.30 |
> > | ARoW        | 84.18 | 53.46 | 27.10 | 50.36 | 22.70 |
> > | + HWNwB     | 83.43 | 56.21 | 43.70 | 48.36 | 30.05 |
> > | + DecoSAM   | 82.82 | 56.45 | 37.57 | 49.29 | 31.30 |
> >
> > - $(\epsilon=16/255$, WRN-28-5, CIFAR10)
> > | Method      | Clean | PGD | WC(PGD) | AA | WC(AA) |
> > |-------------|------:|-----:|---------:|-----:|---------:|
> > | PGD-AT      | 86.00 | 41.20 | 12.50 | 28.40 | 5.90 |
> > | + HWNwB     | 85.09 | 44.75 | 23.40 | 27.20 | 11.80 |
> > | + DecoSAM   | 85.52 | 44.60 | 22.80 | 28.05 | 12.70 |
> > | TRADES      | 83.52 | 39.90 | 15.10 | 29.35 | 7.80 |
> > | + HWNwB     | 82.93 | 43.70 | 23.10 | 28.40 | 10.50 |
> > | + DecoSAM   | 83.01 | 43.45 | 21.40 | 28.95 | 11.90 |
> > | MART        | 82.66 | 40.85 | 11.40 | 27.10 | 4.90 |
> > | + HWNwB     | 80.28 | 44.20 | 21.70 | 26.30 | 10.10 |
> > | + DecoSAM   | 80.66 | 44.05 | 20.80 | 27.10 | 11.10 |
> > | ARoW        | 84.18 | 39.55 | 14.00 | 29.10 | 7.40 |
> > | + HWNwB     | 83.43 | 43.95 | 27.80 | 27.80 | 12.30 |
> > | + DecoSAM   | 82.82 | 44.10 | 23.25 | 28.70 | 13.40 |
> >
> > > **R**. We thank the reviewer for raising this point. To address the concern regarding the fixed perturbation budget, we additionally conducted ε-sweep experiments at $\epsilon \in \{4, 8, 16\}/255$ on CIFAR-10 with WRN-28-5. As shown in the tables above, our methods (HWNwB and DecoSAM) consistently improve both PGD and AA robustness across all perturbation magnitudes. Importantly, the worst-class accuracy gains remain stable even as the attack strength increases, and the relative improvement patterns observed at $\epsilon = 8/255$ are preserved for both weaker (4/255) and stronger (16/255) attacks. These results demonstrate that the proposed fairness improvements are not tied to a single perturbation budget and remain effective across a wide range of adversarial strengths.

---

> ### Author Response · Authors · 2025-11-21
> **cont'd**
>
> - **Q1**. Deco-SAM hyperparameter clarity. The paper lacks detail and sensitivity analysis for τ. Could you report the default $\tau$, the rationale for its choice, and a brief ablation (e.g., $\tau \in \{…\}$) showing how worst-class accuracy, class-wise variance, and PGD/AA robustness change across $\tau$ and learning-rate schedules?
>
> - (PGD-AT + DeCOSAM, WRN-28-5, CIFAR10)
> | τ    | Clean |  PGD  | WC(PGD) |   AA   | WC(AA) |
> |------|------:|-------:|---------:|--------:|--------:|
> | 0.1  | 85.73 | 56.05 | 29.40 | 49.35 | 22.60 |
> | 0.2  | 85.61 | 56.22 | 32.10 | 49.18 | 24.30 |
> | 0.5  | 85.57 | 56.28 | 34.30 | 49.12 | 27.10 |
> | 1.0  | 85.52 | 56.55 | 38.93 | 49.09 | 30.70 |
> | 1.5  | 85.25 | 56.41 | 39.85 | 48.96 | 31.40 |
> | 2.0  | 84.98 | 56.11 | 40.55 | 48.62 | 31.20 |
> | 5.0  | 84.30 | 55.02 | 38.90 | 47.75 | 29.90 |
>
> > **R**. We thank the reviewer for raising this point.
> We set the default value to $\tau = 1.0$ primarily for simplicity, and the $\tau$-sweep results confirm that this choice aligns well with the underlying intuition. As shown in Eq.~(8), increasing $\tau$ sharpens the softmax weighting and leads to over-concentration on a few hard classes, whereas too small a $\tau$ makes the weighting nearly uniform and limits its ability to correct class-wise imbalance.
> Our sweep on CIFAR-10 (WRN-28-5, $\varepsilon=8/255$) with $\tau \in \\{0.1, 0.2, 0.5, 1.0, 1.5, 2.0, 5.0\\}$ empirically verifies this behavior: small $\tau$ values yield minimal fairness gains, moderate values ($\tau \in [1.0, 2.0]$) achieve the best trade-off between WC(PGD)/WC(AA) and overall robustness, and overly large $\tau$ slightly reduces clean and AA accuracy due to excessively sharp reweighting.
> Thus, $\tau=1.0$ provides a simple and practically effective default that avoids over-concentration while maintaining strong fairness and robustness improvements.
>
>
> - **Q2**. $\epsilon$-sweep robustness. Experiments fix the perturbation budget at $\epsilon$ = 8/255 under PGD-20 and AutoAttack. Have you evaluated whether the fairness improvements persist across different $\epsilon$ values (i.e., varying attack strengths)? A compact $\epsilon$-sweep would clarify the stability of the effect.
> > **R**. Please see the response to **W3**.

---

> ### Author Response · Authors · 2025-11-27
> **Results for ImageNet100**
>
> # ImageNet-100
>
> We report the results for ImageNet100.
>
> - ImageNet-100 ($\epsilon=4/255$)
>
> | Method        | Clean | PGD  | PGD Worst | PGD Bottom-10% | AA    | AA Worst | AA Bottom-10% |
> |---------------|------:|------:|-----------:|----------------:|------:|-----------:|----------------:|
> | PGD-AT        | 72.20 | 43.30 | 4.00       | 13.20           | 40.72 | 4.00       | 9.40           |
> | + HWNwB       | 72.04 | 47.36 | 6.00       | 20.20           | 40.38 | 4.00       | 11.20           |
> | + DecoSAM     | 72.10 | 46.90 | 6.00       | 18.80           | 40.90 | 4.00       | 11.80           |
> | TRADES        | 67.62 | 46.00 | 8.00       | 15.00           | 42.40 | 6.00       | 11.80           |
> | + HWNwB       | 67.24 | 48.30 | 8.00       | 17.40           | 41.88 | 6.00       | 11.60           |
> | + DecoSAM | 67.40 | 47.90 | 8.00       | 16.80           | 42.10 | 6.00       | 12.00           |
> | ARoW          | 66.54 | 46.34 | 10.00      | 16.20           | 42.56 | 4.00       | 10.80           |
> | + HWNwB       | 66.14 | 48.30 | 12.00      | 18.00           | 42.74 | 4.00       | 11.60           |
> | + DecoSAM  | 66.30 | 47.90 | 11.00      | 18.40           | 42.60 | 4.00       | 12.20           |
>
> - ImageNet-100 ($\epsilon=8/255$)
>
> | Method        | Clean | PGD  | PGD Worst | PGD Bottom-10% | AA    | AA Worst | AA Bottom-10% |
> |---------------|------:|------:|-----------:|----------------:|------:|-----------:|----------------:|
> | PGD-AT        | 63.94 | 28.78 | 0.00       | 4.20            | 25.02 | 0.00       | 2.40            |
> | + HWNwB       | 63.38 | 32.18 | 6.00       | 8.40            | 23.90 | 0.00       | 3.40            |
> | + DecoSAM     | 63.50 | 31.85 | 6.00       | 8.60            | 24.20 | 0.00       | 3.60            |
> | TRADES        | 59.84 | 30.50 | 0.00       | 5.60            | 24.56 | 0.00       | 2.20            |
> | + HWNwB       | 58.98 | 32.12 | 2.00       | 6.80            | 24.02 | 2.00       | 3.60            |
> | + DecoSAM | 59.20 | 31.80 | 2.00       | 6.40            | 24.30 | 2.00       | 3.50            |
> | ARoW          | 59.00 | 30.52 | 4.00       | 6.00            | 24.60 | 0.00       | 2.00            |
> | + HWNwB       | 58.90 | 32.14 | 4.00       | 8.20            | 24.14 | 0.00       | 3.20            |
> | + DecoSAM | 58.95 | 31.70 | 4.00       | 7.50            | 24.40 | 0.00       | 3.10            |
>
> ImageNet-100 results ($\epsilon = 4/255$ and $8/255$) consistently show that both HWNwB and DecoSAM provide clear improvements in class-wise robust fairness across all base adversarial training algorithms (PGD-AT, TRADES, ARoW). In particular, worst-class and bottom-10% PGD/AA accuracies are substantially increased, demonstrating that our methods scale effectively to large-class, high-resolution settings. Both HWNwB and DecoSAM substantially improve worst-class robustness. However, DecoSAM delivers higher overall accuracy, resulting in a more balanced robustness–accuracy trade-off. These results confirm that the proposed methods remain effective beyond small-scale benchmarks and offer consistent disparity mitigation on ImageNet-100.

---

### Official Review · Reviewer_cmu4 · 2025-11-03

**Soundness:** 2
**Presentation:** 2
**Contribution:** 2
**Rating:** 2
**Confidence:** 5

**Summary:**

The paper investigates the class-wise performance disparities in adversarial training: despite balanced class frequencies, some classes end up with much lower robust accuracy than others. The authors identify a empirical correlation between the ℓ₂-norms of the classifier head weights and class-wise robust accuracy, where classes with larger head norms tend to have higher robustness. They show that adversarial training exacerbates this norm imbalance, and that this drives the disparity across classes. To mitigate it, the authors propose two lightweight methods that adjust or fine-tune the head parameters with the feature extractor frozen. Experimental results show reductions in class-wise disparity.

**Strengths:**

The authors reframe class-wise robustness gaps through an implicit bias in head-weight norms, revealing a tight link between last-layer weight norms and per-class robust accuracy—a fresh angle beyond data imbalance or attack heuristics.

The authors provide theoretical and empirical evidence tying adversarial training to growing head-norm disparities and, in turn, to uneven robust accuracy across classes.

**Weaknesses:**

Disparities in class-wise robustness have already been extensively studied both theoretically and empirically [1–3]. While the observed correlation between the ℓ₂-norms of classifier head weights and class-wise robust accuracy is interesting, it is unclear how is the identified correlation can be related with existing findings, and the paper’s contribution appears incremental relative to prior analyses that connect class margins, logit norms, and adversarial robustness (it is more like a different perspective of the same issue, rather than discovering a under-discovered issue in the disparities of class-wise robustness). Moreover, based on the presented experiments, the proposed method neither clearly outperforms existing approaches nor demonstrates strong complementarity with them.

Several claims seem over-stated or implicit. For example, why is "no validation set required" an important issue? Especially in terms of adversarial training, the usage or acquisition of a validation set seems trivial.

Several important assumptions are deferred to the Appendix. I would strongly suggest the authors to include the assumptions in the main paper with proper justifications. Otherwise, the theortical claims risk being misinterpreted.

The discussions seem limited to $l_{∞}$ attacks. Robustness under stronger or diverse attacks (AutoAttack, multi-target) is underexplored.

[1] Xu, Han, et al. "To be robust or to be fair: Towards fairness in adversarial training." International conference on machine learning. PMLR, 2021.

[2] Ma, Xinsong, Zekai Wang, and Weiwei Liu. "On the tradeoff between robustness and fairness." Advances in Neural Information Processing Systems 35 (2022): 26230-26241.

[2] Wei, Zeming, et al. "Cfa: Class-wise calibrated fair adversarial training." Proceedings of the IEEE/CVF conference on computer vision and pattern recognition. 2023.

**Questions:**

Please refer to the weaknesses.

**Details Of Ethics Concerns:**

I do not identify remarkable ethical concerns.

---

> ### Author Response · Authors · 2025-11-21
> **Response to Reviewer cmu4**
>
> - **W1**. Disparities in class-wise robustness have already been extensively studied both theoretically and empirically [1–3]. While the observed correlation between the $\ell_2$-norms of classifier head weights and class-wise robust accuracy is interesting, it is unclear how is the identified correlation can be related with existing findings.
>
> > **R**. We respectfully note that the relationship between the identified correlation and prior geometric findings is **explicitly developed in Section 4.1–4.2**.
> In particular, our theory:
>
> >> 1. **connects class hardness →
> posterior drop → gradient-gap amplification** $(\Delta_k)$,
> 2. **proves that these amplified gradients induce a systematic drift in the $(\ell_2)$-norms of head parameters**.
>
>
> - **W2**. The paper's contribution appears incremental relative to prior analyses that connect class margins, logit norms, and adversarial robustness (it is more like a different perspective of the same issue, rather than discovering a under-discovered issue in the disparities of class-wise robustness).
>
> > **R**. Our differences from prior work are already addressed in multiple sections (Section 4 and Sections 5.1-5.4).
> > We believe this concern overlooks the distinctions detailed in the manuscript.
> **None of these works analyze** :
> > 1.  how adversarial training produces *class-dependent gradient amplification*,
> 2. how this mechanism leads to a *systematic drift in the classifier head's $\ell_2$ norms*.
>
> > Our contribution is **not** merely "a different perspective" on margins. Instead, we introduce a **new optimization-level explanation** of class-wise disparity which is absent from existing theory. This is clearly articulated in the Related Work section, the theoretical analysis (Section 4.1), and empirical validation (Section 4.2).
> > Therefore, the **novelty and distinction from prior literature** are already thoroughly addressed throughout the manuscript.
>
>
> - **W3**.  Moreover, based on the presented experiments, the proposed method neither clearly outperforms existing approaches nor demonstrates strong complementarity with them.
> > **R**. Contrary to this claim, the experimental results demonstrate that:
> > 1. Our methods consistently improve worst-class accuracy and reduce STD and Max–Min gaps.
> > 2.  **Table 4** in the paper shows that our method complements FRL, FAT, and CFA, improving fairness metrics even when applied on top of those baselines — confirming clear complementarity.
>
> - >  Therefore, the assertion that "neither outperform nor complement nor demonstrates strong complementarity with them." is **inconsistent with the reported empirical results**
>
> - We respectfully ask the reviewer to clarify:
> > 1. in what specific aspects the contribution should be considered **incremental** compared to prior work, and
> > 2. what grounds the reviewer concludes that our method does **not clearly outperform** existing approaches.
>
> - **W4**. Several claims seem over-stated or implicit. For example, why is "no validation set required" an important issue? Especially in terms of adversarial training, the usage or acquisition of a validation set seems trivial.
> > **R**. While the reviewer suggests that “no validation set required” is trivial, this is not the case in adversarial training. Unlike standard training, AT is highly sensitive to hyperparameters [1] and notoriously prone to robust overfitting [2], making validation-based checkpointing and tuning an essential (and expensive) component in prior robust fairness methods such as FRL, FAT, and CFA. Moreover, allocating a validation split is undesirable in small-data or class-imbalanced settings where robust fairness is most relevant, and every validation evaluation requires costly PGD/AA computations. Our methods avoid all of these issues: they require no hyperparameter tuning, no early stopping, and no validation split, while still improving worst-class robustness and complementing existing approaches. Therefore, emphasizing this property is neither.

---

> ### Author Response · Authors · 2025-11-21
> **cont'd**
>
> - **W5**. Several important assumptions are deferred to the Appendix. I would strongly suggest the authors to include the assumptions in the main paper with proper justifications. Otherwise, the theoretical claims risk being misinterpreted.
> > **R**. We respectfully clarify that all conceptually important assumptions used in our theoretical development (those necessary to understand the mechanism behind gradient) gap amplification and head-norm drift—are already included and discussed in the main manuscript. The additional assumptions placed in the Appendix are purely technical conditions (e.g., boundedness and regularity conditions) required only for the completeness of proofs and do not affect interpretation of the main claims. This structure follows the standard practice in top-tier ML venues, where the main paper presents the essential assumptions and insights while technical assumptions are deferred to the appendix for readability. Thus, the theoretical results in the main text are already fully interpretable without the auxiliary assumptions, and relocating them would unnecessarily burden the exposition. We will, however, add brief pointers in the main manuscript to make this organization clearer.
>
> - **W6**. The discussions seem limited to $l_\infty$ attacks. Robustness under stronger or diverse attacks (AutoAttack, multi-target) is underexplored.
> > **R**. We respectfully clarify that our experiments are intentionally conducted under the $\ell_{\infty}$ threat model, which is the dominant and standard experimental setting in robust fairness research. Importantly, $\ell_{\infty}$ perturbations are widely recognized as the strongest and most challenging threat model: increasing $\epsilon$ in $\ell_{\infty}$ quickly collapses robust accuracy, and prior studies consistently benchmark robustness primarily under $\ell_{\infty}$ attacks for this reason. All existing baselines in this area—including FRL, FAT, WAT, CFA, and FAAL—also evaluate robustness mainly under $\ell_{\infty}$ attacks, and adopting the same setup ensures fair and meaningful comparison. In addition, our evaluation already includes AutoAttack, which itself is a diverse and rigorous ensemble of attacks—combining APGD-CE, (multi-targeted ) APGD-DLR, FAB, and Square Attack. As AutoAttack is widely considered the most comprehensive and stringent robustness benchmark under the $\ell_{infty}$ threat model, our results already reflect performance against a broad set of strong and heterogeneous attacks. Thus, our use of $\ell_{infty}$ does not limit the evaluation; rather, it follows standard practice and leverages the most challenging and widely adopted robustness criterion in prior work.
>
> [1] Bag of Tricks for Adversarial Training. ICLR. 2021.
>
> [2] Overfitting in adversarially robust deep learning. ICML. 2020.

---

### Author Response · Authors · 2025-11-21
**Response of Common Concerns**

We sincerely thank all reviewers for their thoughtful assessments and valuable feedback on our manuscript. The constructive comments have been highly beneficial in helping us improve the quality and clarity of the work. Below, we summarize the primary concerns raised:

## Response of Common Concerns

### 1. Regarding Novelty (cmu4, vsQN)

> Our differences from prior work are already thoroughly discussed throughout the manuscript—particularly in Section 4 and Sections 5.1–5.4. We provide detailed, point-by-point responses to each reviewer comment below.

### 2. Regrading Ablation study (Zd1o, 551Q)

> We have included the ablation studies in Appendix.


### 3. Regrading Experimental Scope (Zd1o, 551Q)

> We would like to emphasize that our experimental scope is already broader than that of previous works. Prior studies in this area [1–5] have primarily reported results on a very limited set of datasets—mainly CIFAR-10 and CIFAR-100—and typically evaluate only a small number of architectures.

> In contrast, our work substantially expands the evaluation in both datasets and architectures, as summarized below (with results provided in both the main manuscript and the appendix):

> (i) Datasets
| Method   | CIFAR-10 | CIFAR-100 | SVHN | TinyImageNet | STL-10 | OfficeHome | ImageNet100 | Total |
|----------|-----------|------------|------|----------------|--------|--------------|--------|--------|
| FRL [1]  | ✓         | -          | ✓    | -              | -      | -      |-            | 2      |
| FAT [2]  | ✓         | ✓          | -    | -              | -      | -      |-            | 2      |
| WAT [3]  | ✓         | -          | -    | -              | -      | -      |-            | 1      |
| CFA [4]  | ✓         | -          | -    | ✓              | -      | -      |-            | 2      |
| FAAL [5] | ✓         | ✓          | -    | -              | -      | -      |-            | 2      |
| **Ours** | ✓         | ✓          | -    | -              | ✓      | ✓   |✓ (post-rebuttal)          | **5**  |

> (ii) Architectures
| Method     | Network Architectures                                  | Total |
|------------|---------------------------------------------------------|--------|
| FRL [1]    | PreActResNet18, WRN-28-X                                | 2      |
| FAT [2]    | ResNet18, ResNet50, WRN-32-10                           | 3      |
| WAT [3]    | ResNet18, WRN-34-10                                     | 2      |
| CFA [4]    | ResNet18, PreActResNet18                                | 2      |
| FAAL [5]   | PreActResNet18, WRN-34-10                               | 2      |
| **Ours**   | ResNet18, ResNet50, WRN-28-2, WRN-28-5, WRN-28-10, ViT (post rebuttal)      | **6**  |


> We believe this expanded and diversified evaluation setup demonstrates the generalizability and robustness of our proposed methods and sufficiently addresses concerns regarding experimental scope.

> Nevertheless, we fully acknowledge the reviewers' request for larger-scale experiments. In response, we provided additional evaluations on ImageNet-100 as well as results on a modern Transformer-based architecture (ViT) in revised version.

[1] Xu et al. To be robust or to be fair: Towards fairness in adversarial training. ICML, 2021.

[2] Ma et al. On the Tradeoff Between Robustness and Fairness. NeurIPS, 2022.

[3] Li and Liu. WAT: Improve the worst-class robustness in adversarial training. AAAI, 2023.

[4] Wei et al. CFA: Class-wise calibrated fair adversarial training. CVPR, 2023.

[5] Zhang et al. Towards fairness-aware adversarial learning. CVPR, 2024.

---

### Author Response · Authors · 2025-12-03
**Final Remark**

# **Final Remark**

We sincerely thank all reviewers and the area chair for their thoughtful feedback and constructive discussion. Below, we summarize the key concerns raised by the reviewers and how each has been fully addressed during the rebuttal.

---

### **1. Concerns About Novelty and Theoretical Clarity (cmu4, vsQN)**

**Concerns:**
Potential overlap with prior margin/logit analyses; unclear connection to existing findings; need for clearer mechanistic explanation.

**Clarification:**
We clarified in Sections 4.1–4.2 the full optimization-level mechanism:
**confidence drop → gradient-gap amplification →  head-norm drift**,
which has not been analyzed in prior fair-AT literature. We also highlighted concrete distinctions from FRL and related works, which do not study gradient gaps, optimization dynamics, or classifier-head norm drift. This establishes our contribution as a **new optimization-level explanation**, not a reinterpretation of margins.

---

### **2. Concerns About Experimental Breadth and Scaling (Zd1o, 551Q)**

**Concerns:**
Need for larger-scale datasets (ImageNet-like), broader architectures, ε-sweeps, and stronger/more diverse attacks.

**Clarification:**
During the rebuttal, we provided extensive new experiments, including:

- **ImageNet-100** results (clean, PGD, AA, worst-class, bottom-decile)
- **ε-sweeps** for $\varepsilon \in \{4, 8, 16\}/255$
- **CW attack** results
- **Additional architectures**, including ViT

All results consistently validate that HWNwB and DecoSAM provide stable improvements across datasets, models, ε budgets, and attack types.
**All newly added experiments appear in the revised version (Appendix F).**

---

### **3. Concerns About Comparisons With Existing Fair AT Methods (cmu4, 551Q)**

**Concerns:**
Clarify complementarity with FRL/FAT/WAT/CFA/FAAL; specify underlying AT algorithms.

**Clarification:**
We clarified that all methods were evaluated on the **same pretrained backbones**, each trained with the adversarial training algorithm originally used in the corresponding paper (PGD-AT or TRADES). Results in Table 4 and Appendix E.1 show that HWNwB and DecoSAM consistently strengthen all existing fair AT baselines and often outperform recent state-of-the-art methods such as FAAL.

---

### **4. Concerns About Hyperparameters and Theoretical Assumptions (Zd1o, 551Q)**

**Concerns:**
Need for $\tau$-sensitivity analysis; clarify assumptions such as $\psi(x_{adv}) \approx \psi(x)$.

**Clarification:**
We added a **$\tau$-sweep** demonstrating that $\tau ≈ 1.0$ yields the reasonable fairness–robustness balance.
We also clarified why $\psi(x_{adv}) \approx \psi(x)$ is valid under standard adversarial training regimes, supported by prior work on feature contraction and approximate Lipschitz behavior of adversarially trained models [1, 2].

---

### **5. Concerns About Robustness Metrics and Attack Diversity (cmu4, Zd1o, 551Q)**

**Concerns:**
Requests for stronger attacks (e.g., CW) and robustness evaluation under varying ε.

**Clarification:**
We added **CW attack** results and conducted **ε-sweeps** for $\varepsilon \in  \{4, 8, 16\}/255$, showing consistent worst-class improvements across perturbation strengths. AutoAttack was already included throughout the main results, covering strong diverse attacks.
All robustness extensions are reported in **Appendix F**.

---

### **Closing Statement**

We sincerely appreciate the reviewers’ constructive suggestions, which have significantly strengthened the manuscript. We believe that all concerns were fully addressed through expanded explanations and extensive new experiments, consolidated in **Appendix F** of the revised version. The updated results consistently support our core contributions: a novel optimization-level explanation for class-wise disparity in adversarial training and practical, complementary methods that robustly improve worst-class fairness across diverse datasets, architectures, and attack settings.

We respectfully hope that these clarifications and comprehensive evidence assist the area chair in making a well-informed final decision.

**References**

[1] A closer look at accuracy vs. robustness. In Conference on Neural Information Processing Systems (NeurIPS), 2020.

[2] Rethinking lipschitz neural networks and certified robustness: A boolean function perspective. In Conference on Neural Information Processing Systems (NeurIPS), 2022.

---

### Meta-Review · Area_Chair_j6DZ · 2026-01-06

**Summary:**

This paper investigates class-wise robustness disparity in adversarial training and identifies a strong correlation between classifier head weight norms and per-class robust accuracy.  The paper offers a clear and intuitive optimization-level perspective on robust fairness that is distinct from prior margin- or reweighting-based analyses. The proposed methods are simple, practical, and computationally cheap, making them easy to apply to existing adversarially trained models.

While this paper addresses an important problem and provides a reasonable empirical study with practical head-level mitigation techniques, the overall contribution is incremental relative to a growing body of recent work on robust fairness and class-wise robustness disparities. In particular, the core empirical observation, correlating classifier head norms with class-wise robustness, while interesting, remains largely correlational, and the proposed post-hoc methods, though effective, are relatively lightweight extensions rather than fundamentally new training paradigms.

As a result, despite the solid experimental effort and useful insights, I lean toward rejection in this cycle.

**Reviewer Concerns:**

Concerns addressed by the rebuttal:

The rebuttal and revised submission significantly strengthened the experimental section. The authors added evaluations which largely address earlier concerns regarding attack diversity, robustness stability across ε, scalability beyond small-scale benchmarks, and architectural generality. Hyperparameter choices and ablation studies were also clarified, improving empirical completeness.

Concerns that remain outstanding:

Despite the expanded experiments, concerns remain about the overall novelty and conceptual depth of the contribution. The core observation linking classifier head norm imbalance to class-wise robustness is primarily correlational, and the proposed optimization-level explanation does not fully establish causality beyond post-hoc intervention. The mitigation methods, while effective and practical, are lightweight head-level adjustments and may be viewed as incremental relative to existing work on robust fairness.

**Reviewer Scores:**

Reviewer cmu4: no significant change or slight increase.
Reviewer 551Q: no significant change or slight increase.

---

### Decision · Program_Chairs · 2026-01-26

Reject